

# Global Greenhouse Gas Reconciliation 2022

Zhu Deng[1,2,3], Philippe Ciais[4,*], Liting Hu[5], Adrien Martinez[4], Marielle Saunois[4], Rona L. Thompson[6], Kushal Tibrewal[4], Wouter Peters[7,8], Brendan Byrne[9], Giacomo Grassi[10], Paul I. Palmer[11,12], Ingrid T. Luijkx[7], Zhu Liu[1,2,3,*], Junjie Liu[9,13], Xuekun Fang[5], Tengjiao Wang[14], Hanqin Tian[15], Katsumasa Tanaka[4,16], Ana Bastos[17], Stephen Sitch[18], Benjamin Poulter[19], Clément Albergel[20], Aki Tsuruta[21], Shamil Maksyutov[16], Rajesh Janardanan[16], Yosuke Niwa[16,22], Bo Zheng[23,24], Joël Thanwerdas[25], Dmitry Belikov[26], Arjo Segers[27], Frédéric Chevallier[4]

[1]Department of Geography, University of Hong Kong, Hong Kong SAR, China
[2]Institute for Climate and Carbon Neutrality, University of Hong Kong, Hong Kong SAR, China
[3]Department of Earth System Science, Tsinghua Unverisity, Beijing, China
[4]Laboratoire des Sciences du Climat et de l'Environnement, IPSL, CEA-CNRS-UVSQ, Université Paris-Saclay, Gif-sur-Yvette, France
[5]College of Environmental & Resource Sciences, Zhejiang University, Hangzhou, Zhejiang, China
[6]Norwegian Institute for Air Research (NILU), Kjeller, Norway
[7]Meteorology and Air Quality Department, Wageningen University & Research, Wageningen, the Netherlands
[8]Energy and Sustainability Research Institute Groningen, University of Groningen, Groningen, the Netherlands
[9]Jet Propulsion Laboratory, California Institute of Technology, Pasadena, CA, USA
[10]Joint Research Centre, European Commission, Ispra (VA), Italy
[11]National Centre for Earth Observation, University of Edinburgh, Edinburgh, UK
[12]School of GeoSciences, University of Edinburgh, Edinburgh, UK
[13]Division of Geological and Planetary Sciences, California Institute of Technology, Pasadena, CA, USA
[14]Institute of Blue and Green Development, Shandong University, Weihai, China
[15]International Center for Climate and Global Change Research, School of Forestry and Wildlife Sciences, Auburn University, Auburn, AL 36849, USA
[16]Earth System Division, National Institute for Environmental Studies, Onogawa 16-2, Tsukuba, Ibaraki 305-8506, Japan
[17]Department of Biogeochemical Integration, Max Planck Institute for Biogeochemistry, Hans Knöll Str. 10, Jena, Germany
[18]Faculty of Environment, Science and Economy, University of Exeter, Exeter, UK
[19]NASA Goddard Space Flight Center, Biospheric Sciences Laboratory, Greenbelt, MD 20771, USA
[20]European Space Agency Climate Office, ECSAT, Harwell Campus, Didcot, Oxfordshire, UK
[21]Finnish Meteorological Institute, P.O. Box 503, 00101, Helsinki, Finland
[22]Department of Climate and Geochemistry Research, Meteorological Research Institute (MRI), Nagamine 1-1, Tsukuba, Ibaraki 305-0052, Japan
[23]Shenzhen Key Laboratory of Ecological Remediation and Carbon Sequestration, Institute of Environment and Ecology, Tsinghua Shenzhen International Graduate School, Tsinghua University, Shenzhen, 518055, China
[24]State Environmental Protection Key Laboratory of Sources and Control of Air Pollution Complex, Beijing 100084, China
[25]Empa, Swiss Federal Laboratories for Materials Science and Technology, Dübendorf, Switzerland
[26]Center for Environmental Remote Sensing, Chiba University, Chiba, Japan
[27]TNO, Department of Air quality and Emissions Research, P.O. Box 80015, NL-3508-TA, Utrecht, the Netherland

*Correspondence to*: Philippe Ciais (philippe.ciais@lsce.ipsl.fr); Zhu Liu (zhuliu@hku.hk)

**Abstract.** In this study, we provide an update of the methodology and data used by Deng et al. (2022) to compare the national greenhouse gas inventories (NGHGIs) and atmospheric inversion model ensembles contributed by international research teams coordinated by the Global Carbon Project. The comparison framework uses transparent processing of the net ecosystem



exchange fluxes of carbon dioxide ($CO_2$) from inversions to provide estimates of terrestrial carbon stock changes over managed land that can be used to evaluate NGHGIs. For methane ($CH_4$), and nitrous oxide ($N_2O$), we separate anthropogenic emissions

from natural sources based directly on the inversion results, to make them compatible with NGHGIs. Our global harmonized NGHGIs database was updated with inventory data until February 2023 by compiling data from periodical UNFCCC inventories by Annex I countries and sporadic and less detailed emissions reports by non-Annex I countries given by National Communications and Biennial Update Reports. For the inversion data, we used an ensemble of 22 global inversions produced for the most recent assessments of the global budgets of $CO_2$, $CH_4$ and $N_2O$ coordinated by the Global Carbon Project with

ancillary data. The $CO_2$ inversion ensemble in this study goes through 2021, building on our previous report from 1990 to 2019, and includes three new satellite inversions compared to the previous study, and an improved managed land mask. As a result, although significant differences exist between the $CO_2$ inversion estimates, both satellite and in-situ inversions over managed lands indicate that Russia and Canada had a larger land carbon sink in recent years than reported in their NGHGIs, while the NGHGIs reported a significant upward trend of carbon sink in Russia but a downward trend in Canada. For $CH_4$ and

$N_2O$, the results of the new inversion ensembles are extended to 2020. Rapid increases in anthropogenic CH4 emissions were observed in developing countries, with varying levels of agreement between NGHGIs and inversion results, while developed countries showed a slow declining or stable trend in emissions. Much denser sampling and higher atmospheric $CO_2$ and $CH_4$ concentrations by different satellites, are expected in the coming years. The methodology proposed here to compare inversion results with NGHGIs can be applied regularly for monitoring the effectiveness of mitigation policy and progress by countries

to meet the objective of their pledges. The dataset constructed for this study is publicly available at https://doi.org/10.5281/zenodo.10841716 (Deng et al., 2024).

## 1 Introduction

If modeled pathways align with Nationally Determined Contributions (NDCs) declared prior to COP26 (in 2021) until 2030 and do not involve any subsequent increase in ambition, the projected global warming by 2100 would be 2.1-3.4°C (IPCC,

2023). The global stocktake coordinated by the secretariat of the United Nations Framework Convention on Climate Change (UNFCCC) considers data from national greenhouse gas inventories (NGHGIs) to assess the collective climate progress to curb emissions. It is expected there will be differences in the quality of NGHGIs being reported to the UNFCCC (Perugini et al., 2021). UNFCCC Annex I Parties, which include all OECD (Organisation for Economic Co-operation and Development) countries and several EIT (Economies In Transition) already report annually their emissions following the same IPCC

guidelines (IPCC 2006) in a common reporting format, with a time latency of roughly 1.5 years. In contrast, non-Annex I Parties, mostly developing and less developed countries, are currently not required to provide reports as regularly and as detailed as Annex I Parties and in a few cases use different IPCC Guidelines in their National Communications (NC) or Biennial Update Reports (BUR) submitted to the UNFCCC. Non-Annex I Parties are scheduled in 2024 to move to regular





and harmonized reporting of their emissions in the national inventory reports (NIRs) in the format of common reporting tables

(CRTs), following the Paris Agreement's enhanced transparency framework (ETF).

The IPCC guidelines for NGHGIs encourage countries to use independent information to verify emissions and removals (IPCC, 1997, 2006, 2019), such as comparisons with independently compiled inventory databases (e.g. IEA, CDIAC, EDGAR, FAOSTAT), or with atmospheric mole fraction measurements interpreted by atmospheric inversion models (see Section 6.10.2 in IPCC (2019)). Such verification of 'bottom-up' national reports against 'top-down' atmospheric inversion results is not

mandatory. However, a few countries (e.g. Switzerland, United Kingdom, New Zealand, and Australia) have already added inversions as a consistency check of their national reports. In our study, we utilized the latest global inversion results from the budget assessments of $CO_2$, $CH_4$, and $N_2O$ conducted by the Global Carbon Project (GCP), focusing on three ensembles of inversions with global coverage. Compared to our previous study (Deng et al., 2022), the $CO_2$ inversion ensemble used in this study has been updated to the global $CO_2$ budget of Friedlingstein et al. (2022) that includes nine $CO_2$ inversions using mole

fraction data from the surface network and/or retrieval products from the Greenhouse Gases Observing Satellite (GOSAT) and Orbiting Carbon Observatory-2 (OCO-2) satellites. The $CH_4$ inversion ensemble and $N_2O$ inversion (Tian et al., 2023) ensemble used in this study are also extended to the 2020. As a result, the new ensembles cover up to 2021 for $CO_2$, 2020 for $CH_4$ and 2020 for $N_2O$, compared to 2019, 2017 and 2016 respectively in our previous study (Deng et al., 2022), allowing us to track and analyze the most recent flux variations.

Our framework to process inversion aims at making them comparable to inventories at countries or groups of countries scale (ie,with an area larger than the spatial resolution of atmospheric transport models typically used for inversions). Atmosphericnversions use *a priori* information for the spatial and temporal patterns of fluxes. Some inversions correct prior fluxes at the spatial resolution of their transport models to match atmospheric observations and use spatial error correlations (usually e-folding length scales) that tie the adjustment of fluxes from one grid cell to its neighbors at distances of tens to

hundreds of kilometers. Other inversions adjust fluxes over coarse regions that are larger than the resolution of the transport model, implicitly assuming a perfect correlation of flux errors within these regions, causing an aggregation error (Kaminski et al., 2001). Thus, to minimize aggregation errors, the results of inversions are shown preferentially for selected large area emitter countries or large absorbers in the case of $CO_2$. We have selected a different set of countries/groups of countries for each gas, according to their importance in the global emission budget. According to the median of inversion data we used in

this study, selected countries collectively represent ~70% of global fossil fuel $CO_2$ emissions, ~90% of global land $CO_2$ sink, ～60% of anthropogenic $CH_4$ emissions, and ~55% of anthropogenic $N_2O$ emissions. To more robustly interpret global inversion results for comparison with inventories, we follow the same criterion and choose high-emitting countries covered (if possible) by atmospheric measurements, although most selected tropical countries have few or no atmospheric in-situ stations. Uncertainties are given by the spread among inversion models (min-max range given the small number of inversions), and the

causes for discrepancies with inventories are analyzed systematically and on a case-by-case basis, considering both individual countries and specific greenhouse gases, for annual variations and for mean budgets over several years.





Based on the newly updated inversion results and inventory, and an improvement in the methodology framework proposed in the previous study (Deng et al., 2022), we specifically address the following questions: 1) how do inversion models compare with NGHGIs for the three gases?; 2) what are the plausible reasons for mismatches between inversions and NGHGIs?; and in particular, did the new maps of managed land masks in this study reduce the mismatch between the inversions and NGHGIs for $CO_2$ and $N_2O$?; 3) what independent information can be extracted from inversions to evaluate the mean values or the trends of greenhouse gas emissions and removals?; and does this information exhibit a good agreement with NGHGIs?; and 4) how do satellite-retrieval driven inversion models differ from the surface in-situ and flask sampling driven inversion model results? Sections 2 presents the updated global database of national emissions reports for selected countries and its grouping into sectors, the global atmospheric inversions used for the study, the processing of fluxes from these inversions to make their results as comparable as possible with inventories. The time series of inversions compared with inventories for each gas, with insights on key sectors for $CH_4$ are discussed in **Sections 3 to 5**. The discussion (Section 6) focuses on the plausible reasons for mismatches between inversions and NGHGIs, comparison between inversion ensembles in this study and previous study, and different priors applied in the $CH_4$ inversions. Finally, concluding remarks are drawn on how inversions could be used systematically to support the evaluation and possible improvement of inventories for the Paris Agreement.

## 2 Material and methods

### 2.1 Compilation and harmonization of national inventories reported to the UNFCCC

All UNFCCC Parties shall periodically update and submit their national GHG inventories of emissions by sources and removals by sinks to the Convention parties. Annex I countries submit their NIRs in common reporting format (CRF) tables every year with a complete time series starting in 1990. Non-Annex I Parties are required to submit their NC roughly every four years after entering the Convention and submit BUR, every two years since 2014. Currently, there are in total 427 submissions of NC and over 166 submissions of BUR (UNFCCC, 2021b, a) (**Fig 1**).



**Figure 1. Numbers of non-Annex I parties for each submission round (as of February 28, 2023).** The numbers in the middle of the dots denote the numbers of non-Annex I parties for each submission, while the black dots denote the total number of non-Annex I parties, the blue dots denote the numbers of non-Annex I parties who has submitted National Communications (NC), green dots for Biennial Update Reports (BUR), yellow dots for National Inventory Report (NIR), and purple dots for Technical Annex on REDD+ . The numbers after the NC and BUR denote the total number of submission reports.

We collected NGHGIs data submitted to UNFCCC by February 28, 2023. For Annex I countries, data collection is straightforward, as their reports are provided as Excel files under a Common Reporting Format (CRF) until the year 2020 last accessed on February 28, 2023. For non-Annex I countries, the data were directly extracted from the original reports provided in Portable Document Format (PDF) last accessed on February 28, 2023. Data from successive reports for the same country were extracted, except when they relate to the same years, in which case only the latest version is considered. While Annex I countries are required to compile their inventory following IPCC 2006 guidelines and the subdivision between sectors established by the UNFCCC decision (dec. 24/CP.19), non-Annex I countries are increasingly adopting the IPCC 2006 Guidelines, although some still utilize the older IPCC 1996 Guidelines, with different approaches and sectors. Consequently, the methods used and the reported sectors may differ among NC and BUR reports.

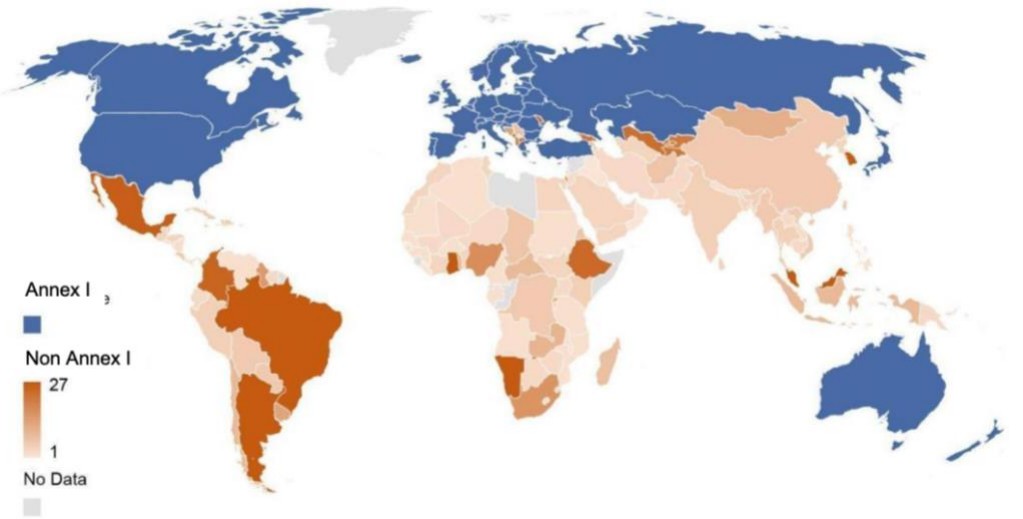

**Figure 2. Number of years covered by NGHGI reports (NC+BUR) in each non-Annex I country (as of February 28, 2023).** Emissions from Greenland are reported by Denmark.

## 2.2 Atmospheric inversions

### CO$_2$ inversions

Nine CO$_2$ inversion systems from the 2022 Global Carbon Budget of the GCP (Friedlingstein et al., 2022) are used, including CarbonTracker-Europe (CTE) v2022 (van der Laan-Luijkx et al., 2017), Jena Carboscope v2022 (Rödenbeck et al., 2003), the





surface air-sample inversion from the Copernicus Atmosphere Monitoring Service (CAMS) v21r1 (Chevallier et al., 2005),
the inversion from the CAMS Satellite FT21r2 (Chevallier et al., 2005), the inversion from the University of Edinburgh (UoE)
v6.1b (Feng et al., 2016), the NICAM-based Inverse Simulation for Monitoring $CO_2$ (NISMON-$CO_2$) v2022.1 (Niwa et al.,
2022), CMS-Flux v2022 (Liu et al., 2021), GONGGA v2022 (Jin et al., 2023), and THU v2022 (Kong et al., 2022). A variety
of transport models are used by these systems, which allows for representing a major driver factor behind differences in flux
estimates based on atmospheric inversions, particularly their distribution over latitudinal bands. Among the nine inversions,
four systems (CAMS Satellite FT21r2, GONGGA v2022, THU v2022, and CMS-Flux v2022) utilize satellite $CO_2$ column
retrievals from GOSAT and/or OCO-2, calibrated to the World Meteorological Organization (WMO) 2019 standards. CMS-
Flux additionally incorporates in-situ observed $CO_2$ mole fraction records. The remaining five inversion systems (CAMS
v21r1, CTE v2022, Jena Carboscope v2022, UoE v6.1b, and NISMON-CO2 v2022.1) solely rely on $CO_2$ mole fractions that
were observed in-situ or collected in flasks (Schuldt et al., 2021, 2022). The $CO_2$ inversion records extend up to and including
2021. Their flux estimates are available at https://meta.icos-cp.eu/objects/GahdRITjT22GGmq_GCi4o_wy and details are
summarized in **Table 1**.

**Table 1 | Atmospheric CO₂ inversions used in this study** (Friedlingstein et al., 2022)

| Inversion System | Version | Period | Observation |
|---|---|---|---|
| CarbonTracker Europe (CTE): CTE2022_SiB4 (van der Laan-Luijkx et al., 2017) | v2022 | 2001-2021 | |
| Jena Carboscope sEXTocNEET (Rödenbeck et al., 2003) | v2022 | 1960-2021 | |
| Copernicus Atmosphere Monitoring Service (CAMS) (Chevallier et al., 2005) | v21r1 | 1979-2021 | Ground-based Obspack GLOBALVIEW plus v7.0 and NRT_v7.2 |
| The University of Edinburgh (UoE) (Feng et al., 2016) | v6.1b | 2001-2021 | |
| the NICAM-based Inverse Simulation for Monitoring $CO_2$ (NISMON-CO2) (Niwa et al., 2022) | v2022.1 | 1990-2021 | |
| CMS-Flux (Liu et al., 2021), | v2022 | 2010-2021 | Ground-based & ACOS-GOSAT v9r; OCO-2 v10 scaled to WMO2019 |
| CAMS-Satellite (Chevallier et al., 2005) | FT21r2 | 2010-2021 | bias-corrected ACOS GOSAT v9 over land until August 2014 + bias- corrected ACO S OCO-2 v10 over land, both rescaled to WMO2019 |
| THU (Kong et al., 2022) | v2022 | 2015-2021 | OCO-2 v10r data scaled to WMO2019 |
| GONGGA (Jin et al., 2023) | v2022 | 2015-2021 | OCO-2 v10r data scaled to WMO2019 |



## CH$_4$ inversions

The CH$_4$ emissions come from the new ensemble of inversions (Saunois et al. in prep.) from 2000 to 2020, using seven different inverse systems for a total nine inversions (**Table 2**). The inverse systems include: CarbonTracker-Europe CH4 (Tsuruta et al., 2017), LMDZ-PYVAR (Yin et al., 2015; Zheng et al., 2018), CIF-LMDZ(Berchet et al., 2021), MIROC4-ACTM (Patra et al., 2018; Chandra et al., 2021), NISMON-CH4 (Niwa et al., 2022), NIES-TM-FLEXPART (Wang et al., 2019; Maksyutov et al., 2021), and TM5-CAMS (Segers and Houweling, 2017). This ensemble of inversions gathers various chemistry transport

models, differing in vertical and horizontal resolutions, meteorological forcing, advection and convection schemes, and boundary layer mixing. Including these different systems is a conservative approach that allows to cover different potential uncertainties of the inversion, among them: model transport, set-up issues, and prior dependency. All inversions except two, use updated common prior emission maps for natural and anthropogenic prior emissions divided into 12 sectors, particularly the EDGAR v6 inventory for prior fossil fuel emissions (Crippa et al., 2021a extrapolated to Jan 1st, 2021), GFED for fires

and ecosystem models for wetland emissions. During the production of the inversion simulations, it was proposed to use another prior for fossil fuel sources, GAINS inventory (REF) instead of . GAINS has higher fossil emissions, in particular over the US and a higher increase of fossil emissions over time in the US (Tibrewal et al., 2024). As Tibrewal et al. showed that inversions are strongly attracted to their priors, comparison between results with GAINS and EDGAR v6 priors is informative about how robust are inversions to their priors when they are used to 'verify' NGHGIs.

Some inversions optimize emissions in groups of sectors, and others only provide total gridded emissions. For the latter, we computed the emission from each sector within each pixel based on the proportion of the prior fluxes. The inversions assimilating surface stations mole fraction observations provide results since 2000, and those assimilating satellite observations from column CH$_4$ measurements (XCH$_4$) of the GOSAT satellite provide results since 2010, first full year of GOSATobservations. Inversion results were gridded into 1° by 1° monthly emission maps and aggregated nationally using a

country mask (Klein Goldewijk et al., 2017).

**Table 2 | Atmospheric CH$_4$ inversions used in this study (Saunois et al. in prep.)**

| Inversion system | Abbreviation | Institution | Observations | Period |
|---|---|---|---|---|
| Carbon Tracker-Europe CH4 | CTE | FMI | Surface stations | 2000-2020 |
| CIF-LMDz | CIF-LMDz | LSCE/CEA | Surface stations | 2000-2020 |
| LMDz-PYVAR | PYVAR-LMDz | LSCE/CEA | GOSAT Leicester v7.2 | 2010-2020 |
| MIROC4-ACTM | MIROC4-ACTM | JAMSTEC | Surface stations | 2000-2020 |
| NISMON-CH4 | NISMON-CH4 | NIES/MRI | Surface stations | 2000-2020 |
| NIES-TM-FLEXPART (NTF) | NIES | NIES | Surface stations | 2000-2020 |
| NIES-TM-FLEXPART (NTF) | NIES | NIES | Surface + GOSAT NIES L2 v02.95 | 2010-2020 |





| TM5-CAMS | TM5 | TNO/VU | Surface stations | 2000-2020 |
| TM5-CAMS | TM5 | TNO/VU | GOSAT ESA/CCI v2.3.8 (combined with surface observations) | 2010-2020 |

**N₂O inversions**

Four N$_2$O inversion systems from the updated GCP Nitrous Oxide Budget (Tian et al., 2023) are used: INVICAT (Wilson et al., 2014), PyVAR-CAMS (Thompson et al., 2014), MIROC4-ACTM (Patra et al., 2018, 2022) and GEOS-Chem (Wells et al., 2015). The N$_2$O inversion results are updated up to 2020.

**Table 3 | Atmospheric N₂O inversions used in this study** (Tian et al., 2023)

| Inversion system | Institution | Period |
| --- | --- | --- |
| INVICAT (Wilson et al., 2014) | Univ. Leeds | 1995-2020 |
| PyVAR-CAMS (Thompson et al., 2014), | NILU/LSCE | 1995-2020 |
| MIROC4-ACTM (Patra et al., 2018, 2022) | JAMSTEC | 1997-2019 |
| GEOS-Chem (Wells et al., 2015) | Univ. Minnesota | 1995-2019 |

**Aggregating the gridded inversion results into national totals**

To obtain national annual-scale flux estimates, we aggregated the gridded flux maps of each inversion with various native resolutions following the methodology outlined in Chevallier (2021). This involved using the 0.08° x 0.08° land country mask of Klein Goldewijk et al. (2017) to calculate the fraction of each country in each inversion grid box.

**2.3 Processing of CO₂ inversion data for comparison with NGHGIs**

**Fossil fuel emissions re-gridding - managed land mask**

To analyze terrestrial CO$_2$ fluxes, we subtracted the same fossil fuel emissions (including cement) of GridFEDv2022.2 (Jones et al., 2022) from the total CO$_2$ flux of each inversion. This is equivalent to assuming perfect knowledge of fossil emissions, adding up to a global total of 9.7 GtC/yr for the year 2021. The dataset used national annual emissions estimates from the 2022 global carbon budget (Friedlingstein et al., 2022) which uses the reported NGHGIs data from Annex I countries and are assumed to be broadly consistent with the non-Annex I countries. This assumption may lead to underestimating the uncertainty of terrestrial CO$_2$ fluxes deduced from inversions.

As defined in the IPCC Guidelines for NGHGIs (IPCC, 2006), only CO$_2$ emissions and removals from managed land are reported in NGHGIs as a proxy for human-induced effects (direct effects and indirect effects such as CO$_2$ fertilization and nitrogen deposition). However, inversion models retrieve all CO$_2$ fluxes (due to both direct and indirect effects, plus the natural





interannual variability) over all lands. We thus retained inversions' national estimates of the Net Ecosystem Exchange (NEE) $CO_2$ flux ($F_{ML}^{inv\ NEE}$) over managed lands grid cells only ($ML$, here defined as all land except intact forests) because the fluxes over unmanaged land are not counted by NGHGIs. We use NEE from the definition of Ciais et al. (2020), representing all non-

fossil $CO_2$ exchange fluxes between terrestrial surfaces and the atmosphere. Other work may use Net Biome Production (NBP) with a similar meaning. $CO_2$ fluxes over unmanaged lands were excluded from the terrestrial $CO_2$ flux totals that will be compared with NGHGIs, proportional to their presence in each inversion grid box. The new maps of non-intact forests are compiled by Grassi et al. (2023). These maps include official country-managed forest and other managed land areas for Canada and Brazil used for their NGHGIs, and the intact forest map (Potapov et al., 2017) as a substitute for unmanaged land where

country-based information is not available. For Russia, we used non-intact forest maps for each province with thresholds adjusted to match the official managed land areas from Russia's NIRs, and assumed that all grasslands were managed. This approach assumes that non-intact forest areas can serve as a reasonably good proxy for managed forests reported in the NGHGIs (Grassi et al., 2021, 2023). It is important to note that this approach is somewhat arbitrary, as highlighted in previous studies (Ogle et al., 2018; Chevallier, 2021; Grassi et al., 2021). However, in the absence of a machine-readable definition of

managed plots in many NGHGIs, there is currently no better alternative available.

**Adjusting $CO_2$ fluxes due to lateral carbon transport by crop and wood products trade and by rivers**

In addition to the extraction of managed land $CO_2$ flux, there are $CO_2$ fluxes that are part of $F_{ML}^{inv\ NEE}$ but are not counted by NGHGIs. These fluxes are induced by (i) soils to rivers to oceans carbon export ($F_{ML}^{rivers}$) which has an anthropogenic and a natural component (Regnier et al., 2013), and (ii) net anthropogenic export of crop and wood products across each country's

boundary ($F_{ant}^{crop\ trade}$ and $F_{ant}^{wood\ trade}$). The magnitudes of these $CO_2$ fluxes are different between countries, and values from the selected countries are presented in **SI Fig 1**. We assume that NGHGIs include $CO_2$ losses from fire (wildfire and prescribed fire) and other disturbances (wind, pests) and from domestic harvesting, as recommended by the IPCC reporting guidelines (IPCC, 2006, 2019) (although some countries, such as Canada and Australia exclude some emissions from these disturbances, and the subsequent removals from the same areas (Grassi et al., 2023)). The adjusted inversion NEE that can be compared

with inventories, $F_{adj}^{inv\ NEE}$, is given by:

$$F_{adj}^{inv\ NEE}= F_{ML}^{inv\ NEE} - F_{ML}^{rivers} - F_{ant}^{crop\ trade} - F_{ant}^{wood\ trade} \quad \Leftrightarrow \quad F_{ant}^{ni}, \qquad (1)$$

where the sign $\Leftrightarrow$ means 'compared with', $F_{ant}^{ni}$ is the anthropogenic $CO_2$ flux from NGHGIs, $F_{tot}^{rivers}$ is the sum of the natural and anthropogenic $CO_2$ flux on land from $CO_2$ fixation by plants that is leached as carbon via soils and channeled to inland waters to be exported to the ocean or to another country. All countries export river carbon, but some countries also receive

river inputs, e.g., Romania receives carbon from Serbia via the Danube River. We estimated the lateral carbon export by rivers minus the imports from rivers entering each country, including dissolved organic carbon, particulate organic carbon and dissolved inorganic carbon of atmospheric origin distinguished from lithogenic, by using the data and methodology described by Ciais et al. (2021). Data are from Mayorga et al. (2010) and Hartmann et al. (2009) and follow the approach of Ciais et al.



(2021) proposed for large regions. We also extracted the lateral flux by rivers over the managed land by using the same methodology as inversion $CO_2$ flux. Thus, in a country that only exports river carbon to the ocean, the amount of carbon exported is equivalent to an atmospheric $CO_2$ sink, denoted as $F_{ML}^{rivers}$ as in eq. (1), thus ignoring burial, which is a small term. Over a country that receives carbon from rivers flowing into its territory, a small national $CO_2$ outgassing is produced by a fraction of this imported flux. In that case, we assumed that the fraction of outgassed to incoming river carbon is equal to the fraction of outgassed to soil-leached carbon in the RECCAP2 region to which a country belongs, estimated with data from

Ciais et al. (2021).

$F_{ant}^{crop\ trade}$ is the sum of $CO_2$ sinks and sources induced by the trade of crop products. This flux was estimated from the annual trade balance of crop commodities calculated for each country from data from the United Nations Statistics Division of the Food and Agriculture Organization (FAOSTAT) combined with the carbon content values of each commodity (Xu et al., 2021). All the traded carbon in crop commodities is assumed to be oxidized as $CO_2$ in one year, neglecting stock changes of

products, and the fraction of carbon from crop products going to waste pools and sewage waters after consumption, thus not necessarily oxidized to atmospheric $CO_2$. $F_{ant}^{wood\ trade}$ is the sum of $CO_2$ sinks and sources induced by the trade of wood products (Zscheischler et al., 2017). Here, we followed Ciais et al. (2021) who used a bookkeeping model to calculate the fraction of domestically produced and imported carbon in wood products that are oxidized in each country during subsequent years, with product lifetimes defined by Mason Earles et al (2012). The underlying assumption in estimating CO2 fluxes from

wood harvest is that the emissions from domestically harvested wood, in addition to imported wood minus exported wood that is not allocated to wood product pools, are released into the atmosphere during the year of harvest. Conversely, wood allocated to wood product pools is gradually released into the atmosphere over time, based on their respective lifetimes. Domestic harvest is assumed to be balanced by an atmospheric $CO_2$ sink of equivalent magnitude, which is not necessarily the case given that harvest is rarely in equilibrium with forest increment, but inversions NEE will correct for this imbalance in our results,

and can thus be compared with NGHGIs. We included in the $F_{ant}^{crop\ trade}$ flux the emissions of $CO_2$ by domestic animals consuming specific crop products delivered as feed. On the other hand, emissions of $CO_2$ from grazing animals and the decomposition of their manure are supposed to occur in the same grid box where grass is grazed, so that the $CO_2$ net flux captured by an inversion is comparable with grazed grasslands' carbon stock changes of inventories. Emissions of reduced carbon compounds (VOCs, $CH_4$, CO) are not included in this analysis (see Ciais et al. (2021) for a discussion of their

importance in inversion $CO_2$ budgets).

In summary, the purpose of the adjustment of eq. (1) is to make inversion output comparable to the NGHGIs that do not include $F_{ML}^{rivers}$, $F_{ant}^{crop\ trade}$ and $F_{ant}^{wood\ trade}$. The UNFCCC accounting rules (IPCC, 2006) assume that all the harvested wood products are emitted in the territory of a country that produces them, which is equivalent to ignoring $F_{ant}^{wood\ trade}$ as a national sink or source of $CO_2$, hence the need to remove $F_{ant}^{wood\ trade}$ from inversion NEE. The adjusted inversion fluxes from eq. (1)

depict the national $CO_2$ stock change which match better the carbon accounting system boundaries of UNFCCC NGHGIs. In the following, we will only discuss adjusted inversion $CO_2$ fluxes ($F_{adj}^{inv\ NEE}$), but for simplicity call them "inversion fluxes".



**2.4 Processing of CH₄ inversions for comparison with national inventories**

Most atmospheric inversions derive total net $CH_4$ emissions at the surface as it is difficult for them to disentangle overlapping emissions from different sectors at the pixel/regional scale based on atmospheric $CH_4$ observations only. However, six of the

seven inverse systems solve for some source categories owing to different spatio-temporal distributions between the sectors. For each inversion, monthly gridded posterior flux estimates were provided at 1°x1° grid resolution for the net flux at the surface ($E_{net}^{inv}$), the soil uptake at the surface ($E_{soil}^{inv}$), the total emission at the surface ($E_{tot}^{inv}$) and five emitting 'super sectors' which regroup several IPCC sectors: Agriculture & Waste ($E_{AgW}^{inv}$), Fossil Fuel ($E_{FF}^{inv}$), Biomass & Biofuel Burning ($E_{BB}^{inv}$), Wetlands ($E_{Wet}^{inv}$), and Other Natural ($E_{Oth}^{inv}$) emissions. Considering the soil uptake as a 'negative source' given separately, the

following equations apply:

$$E_{net}^{inv} = E_{tot}^{inv} + E_{soil}^{inv} = E_{AgW}^{inv} + E_{FF}^{inv} + E_{BB}^{inv} + E_{Wet}^{inv} + E_{Oth}^{inv} + E_{soil}^{inv} \qquad (2)$$

For inversions solving for net emissions only, the partition to source sectors was created based on using a fixed ratio of sources calculated from prior flux information at the pixel scale. For inversions solving for some categories, a similar approach was used to partition the solved categories to the five aforementioned emitting sectors. Such processing can lead to significant

uncertainties if not all sources increase or change at the same rate in a given region/pixel. National values have been estimated using the country land mask described in the $CO_2$ section, thus offshore emissions are not counted as part of inversion results unless they are in a coastal grid cell.

In our previous study (Deng et al., 2022), four methods were proposed to separate $CH_4$ anthropogenic emissions from inversions ($E_{Anth}^{inv}$) to compare them with national inventories ($E_{Anth}^{ni}$). The calculations of anthropogenic emissions by each

method were performed separately for GOSAT inversions and in-situ inversions. However, the differences in the calculated results among the four methods were smaller compared to the variations observed in the inversions (see Deng et al. (2022) Fig 9). Therefore, we apply only one method in this study which consists of using inversion partitioning as defined in Saunois et al. (2020):

$$E_{Anth}^{inv} = E_{AgW}^{inv} + E_{FF}^{inv} + E_{BB}^{inv} - E_{wildfires}^{BU} \Leftrightarrow E_{Anth}^{ni} \qquad (3)$$

This method has some uncertainties. First, the partitioning relies on prior fractions within each pixel, and second, emissions from wildfires are counted for in the Biomass and Biofuel burning ($BB$) inversion category while they are not necessarily reported in NGHGIs. The BB inversion category includes methane emissions from wildfires in forests, savannahs, grasslands, peats, agricultural residues, and the burning of biofuels in the residential sector (stoves, boilers, fireplaces). Therefore, we subtracted bottom-up ($BU$) emissions from wildfires ($E_{wildfires}^{BU}$) based on the GFEDv4 dataset (van Wees et al., 2022) using

their reported dry matter burned and $CH_4$ emission factors. Because the GFEDv4 dataset also reports specific agricultural and waste fire emissions data, we assumed that those fires (on managed lands) are reported by NGHGIs, so they were not counted in $E_{wildfires}^{BU}$.



## 2.5 Processing of N$_2$O inversions for comparison with inventories

We subtracted estimates of natural N$_2$O sources from the N$_2$O emission budget ($E_{tot}^{inv}$) of each inversion, to provide inversions

of anthropogenic emissions ($E_{ant}^{inv}$) that can be compared with national inventories ($E_{ant}^{ni}$).

$$E_{ant}^{inv} = E_{ML}^{inv} - E_{nat}^{aq} - E_{wildfires}^{GFED} \Leftrightarrow E_{ant}^{ni} \tag{4}$$

In our previous study, intact forest grid cells (assumed unmanaged) from Potapov et al. (2017) and lightly grazed grassland areas from Chang et al. (2021) were removed from the gridded N$_2$O emissions in proportion to their presence in each inversion grid box. Here we used the new managed land mask defined in **Section 2.3** to filter gridded N$_2$O emissions from inversions

We verified that the inversion grid box fractions classified as unmanaged do not contain point source emissions from the industry, energy, and diffuse emissions from the waste sector, to make sure that we do not inadvertently remove anthropogenic sources by masking unmanaged pixels. From the EDGARv4.3.2 inventory (Janssens-Maenhout et al., 2019), we found that N$_2$O from wastewater handling covers a relatively large area that might be partly located in unmanaged land. But the corresponding emission rates are more than 1 order of magnitude smaller than those from agricultural soils. For other sectors,

only very few of the unmanaged grid boxes contain point sources, and none of them have an emission rate that is comparable with agricultural soils (managed land). Thus, our assumption that emissions from these other anthropogenic sectors are primarily over managed land pixels is solid (other sectors include: the power industry; oil refineries and transformation industry; combustion for manufacturing; aviation; road transportation no resuspension; railways, pipelines, off-road transport; shipping; energy for buildings; chemical processes; solvents and products use; solid waste incineration; wastewater handling;

solid waste landfills).

The flux $E_{nat}^{aq}$ is the natural emission from freshwater systems given by a gridded simulation of the DLEM model (Yao et al., 2019) describing pre-industrial N$_2$O emissions from N leached by soils and lost to the atmosphere by rivers in the absence of anthropogenic perturbations (considered as the average of 1900-1910). Natural emissions from lakes were estimated only at a global scale by Tian et al. (2020), and represent a small fraction of rivers' emissions. Therefore, they are neglected in this

study. The flux $E_{wildfires}^{GFED}$ is based on the GFED4s dataset (van Wees et al., 2022) using their reported dry matter burned and N$_2$O emission factors. Because the GFED dataset reports specific agricultural and waste fire emissions data, we assume that those fires (on managed lands) are reported by NGHGIs so they were not counted in $E_{wildfires}^{GFED}$ just like for CH4 emissions. Note that there could also be a background natural N$_2$O emission from natural soils over managed lands ($E_{managed\ land}^{soil}$) which is not necessarily reported by NGHGIs. We did not try to subtract this flux from managed land emissions because we assumed

that, after a land use change from natural to fertilized agricultural land, background emissions decrease and become very small compared to N-fertilizers induced anthropogenic emissions. In a future study, we could use for $E_{managed\ land}^{soil}$ the estimate given by simulations of pre-industrial N$_2$O emissions from the NMIP ensemble of dynamic vegetation models with carbon-nitrogen interactions (number of models; n = 7). Namely, their simulation S0 in which climate forcing is recycled from 1901-1920; CO$_2$ is at the level of 1860, and no anthropogenic nitrogen is added to terrestrial ecosystems (Tian et al., 2019).



Another important point to ensure a rigorous comparison between inversion and NGHGI data is whether anthropogenic indirect emissions (AIE) of $N_2O$ are reported in NGHGI reports. This is not always the case even though UNFCCC parties are required to report these in their NGHGIs according to the IPCC guidelines. For example, South Africa's BUR3 did not report indirect $N_2O$ emissions due to the lack of activity data. AIE arise from anthropogenic nitrogen from fertilizers leached to rivers and anthropogenic nitrogen deposited from the atmosphere to soils. AIEs represent typically 20% of direct anthropogenic emissions

and cannot be ignored in a comparison with inversions. For Annex I countries, AIEs are systematically reported, generally based on emission factors since these fluxes cannot be directly measured, and we assumed that indirect emissions only occur on managed land. For non-Annex I countries, we checked manually from the original NC and BUR documents if AIE was reported or not by each non-Annex I country. If AIEs were reported by a country, they were used as such to compare NGHGI data with inversion results, and grouped into the agricultural sector. If they were not reported, or if their values were outside

plausible ranges, AIE were independently estimated by the perturbation simulation of N fertilizers leaching, $CO_2$ and climate on rivers and lakes fluxes in the DLEM model (Yao et al., 2019), and by the perturbation simulation of atmospheric nitrogen deposition on $N_2O$ fluxes from the NMIP model ensemble (Tian et al., 2019).

**2.6 Grouping sectors for comparison**

The bottom-up NGHGIs are compiled based on activity data (statistics) following the IPCC 1996/2006 Guidelines (IPCC,

1997, 2006) with detailed information on subsectors. However, the top-down inversions can only distinguish between very few groups of sectors at most. Thus, in this study, we aggregated NGHGI sectors into some 'super sectors' to make inversions and inventories comparable for each GHG (**Table 2**). For $CO_2$, the inversions are divided into two aggregated super-sectors: fossil fuel and cement $CO_2$ emissions, and adjusted net land flux. Inversions use a prior gridded fossil fuel dataset as summarized in **Section 1.2**, thus, in this study, we compare only the net land flux between inversions and inventories. To

calculate the net land flux over managed lands from NGHGIs, we subtracted fossil emissions from the IPCC/CRF *1. Energy* and *2. Industrial Processes* (or *2. Industrial Processes and Product Use*) sectors from the *Total GHG emissions including LULUCF/LUCF* (or *Total national emissions and removals*) sector. For $CH_4$, we compare inversions and inventories based on three super sectors, including *Fossil*, *Agriculture and Waste*, and *Total Anthropogenic*. To compare with NGHGIs, we group the IPCC/CRF sectors of *1. Energy* and *2. Industrial Processes* (or *2. Industrial Processes and Product Use*) by excluding

Biofuel Burning (reported under *1. Energy* sector) into the super sector of *Fossil;* we group sectors of *4. Agriculture* (or *3. Agriculture*) and *6. Waste* (or *5. Waste*) into the super sector of *Agriculture and Waste*; and we aggregate anthropogenic flux from *Fossil* and *Agriculture and Waste* and *Biofuel Burning* into *Anthropogenic*. For $N_2O$, we grouped the NGHGI sectors into *Anthropogenic* flux being the sum of *1. Energy* + *2. Industrial Processes* (or *2. Industrial Processes and Product Use*) + *4. Agriculture* (or *3. Agriculture*) + *6. Waste* (or *5. Waste*) + *Anthropogenic Indirect Emissions*.

**Table 2. Grouping of NGHGIs sectors into aggregated 'super-sectors' for comparisons with inversions.** * Biofuel burning is likely not included in NGHGIs but under *1.A.4 Other Sectors* if it is reported. ** Field burning of agricultural residues is reported in Annex I countries



under the Agricultural sector. Note that indirect $N_2O$ emissions are reported by Annex I countries but not systematically by non-Annex I ones

| Gas | Super-Sectors | Inversions | NGHGIs (IPCC/CRF) |
|---|---|---|---|
| $CO_2$ | *Net Land Flux (adjusted)* | *Total - Fossil - lateral C* | Non-Annex I (IPCC): *Total GHG emissions including LULUCF/LUCF - (Energy + Industrial Processes)* Annex I (CRF): *Total national emissions and removals) - (Energy + Industrial Processes and Product Use)* |
| $CH_4$ | *Anthropogenic* | *Fossil + Agriculture & Waste + Biofuel Burning* | Energy + Industrial Processes + Agriculture + Waste + Biofuel Burning* |
| | *Fossil* | *Fossil* | Energy + Industrial Processes - Biofuel Burning* |
| | *Agriculture and Waste* | *Agriculture & Waste* | Agriculture + Waste - Field burning of agricultural residues** |
| $N_2O$ | *Anthropogenic* | Total - pre-industrial inland waters | Agriculture + Waste direct + anthropogenic indirect emissions (AIE = anthropogenic N leached to inland waters + anthropogenic N deposited from atmosphere) + energy and industry |

## 2.7 Choice of example countries for analysis

For the analysis, we selected 12 countries (or groups of countries) based on specific criteria for each aggregated sector. Firstly, each chosen country had to possess a sufficiently large land area, as the limitations of coarse-spatial-resolution inversions make it difficult to reliably estimate GHG budgets for smaller countries. Additionally, it was preferable for the selected countries to have some coverage provided by the in situ global network of monitoring stations.

For $CO_2$, we focus on the land CO2 fluxes of large fossil fuel $CO_2$ emitters. Although inversions do not allow to verify fossil

emissions in these countries as they are used as a fixed prior map of emissions, it is crucial to compare the magnitude of national land $CO_2$ sinks with fossil fuel $CO_2$ emissions in those large emitters. It is important to note that fitting net fluxes to changes in atmospheric CO2 and then subtracting the prior fossil fuel (FF) fluxes can result in errors in the residual values, which are typically attributed exclusively to the sum of all non-FF fluxes. Additionally, we included two large boreal forested countries (Russia - RUS and Canada - CAN), two tropical countries with large forest areas (Brazil - BRA and the Democratic

Republic of Congo - COD), two large countries with ground-based stations (Mongolia - MNG and Kazakhstan - KAZ), and two large dry Southern Hemisphere countries also with high rankings in fossil fuel $CO_2$ emissions (South Africa - ZAF and Australia - AUS), both of which possess atmospheric stations to constrain their land $CO_2$ flux.

For $CH_4$, we first ranked countries (or groups of countries) based on their total anthropogenic, fossil, and agricultural emissions. This study includes China (CHN), India (IND), the United States (USA), the European Union (EUR), Russia (RUS), Argentina





(ARG) and Indonesia (IND), all of which are among the top emitters of both fossil fuel and agricultural $CH_4$ and possess large areas. Criteria of large land areas and the presence of atmospheric stations is crucial for in situ inversions. The advantage of utilizing GOSAT in CH4 atmospheric inversions is its ability to provide observations over countries where surface in-situ data are sparse or absent, such as in the tropics. This allows us to consider countries with limited or few ground-based observations. Small countries were excluded due to the coarse spatial resolution. However, among the selected countries, Venezuela, with

an area of 916,400 $km^2$, was chosen specifically for the analysis of $CH_4$ emissions. Despite being relatively small, Venezuela is a large emitter of oil and gas, potentially allowing for inversions using GOSAT satellite observations to constrain its emissions. In major oil- and gas-extracting countries that have negligible agricultural and wetland emissions like Kazakhstan (KAZ), grouped in this study with Turkmenistan (TKM) into KAZ&TKM; Iran (IRN); and Persian Gulf countries (GULF), fossil emissions should be easier to separate by inversions and thus to be compared with NGHGIs.

For $N_2O$, we selected the top 12 emitters based on the NGHGIs reports. Anthropogenic $N_2O$ emissions in most of these countries are predominantly driven by the agricultural sector, which accounts for a share (including indirect emissions) ranging from 6% in Venezuela (VEN) to 95% in Brazil (BRA) of their total NGHGIs emissions.

Together, the selected countries (or groups of countries) with a different selection for each gas, account for more than 90% of the global land $CO_2$ sink, 60% of the global anthropogenic $CH_4$ emissions (around 15% of fossil fuel emissions and

approximately 40% of agriculture and waste emissions separately), and 55% of the global anthropogenic $N_2O$ emissions, as estimated by the NGHGIs.

**Table 3. Lists of countries or groups of countries are analyzed and displayed in the result section for each aggregated sector.** Argentina (ARG), Australia (AUS), BRA (Brazil), Bangladesh (BGD), Canada (CAN), China (CHN), Columbia (COL), Democratic Republic of the Congo (COD), Indonesia (IDN), India (IND), Iran (IRN), European Union (EUR), Kazakhstan (KAZ), Mexico (MEX),

Mongolia (MNG), Nigeria (NGA), Pakistan (PAK), Russia (RUS), South Africa (ZAF), Sudan (SDN), Thailand (THA), United States (USA), Venezuela (VEN), GULF = Saudi Arabia + Oman + United Arab Emirates + Kuwait + Bahrain + Iraq + Qatar, KAZ&TKM = Kazakhstan + Turkmenistan. For $CH_4$, acronyms underlined denotes the countries appear in both *Anthorpogenic* and *Fossil* or *Agriculture and Waste* sectors.

| Gas | Super Sector | Country List |
|-----|-------------|-------------|
| $CO_2$ | *Net Land Flux* | AUS, BRA, CAN, CHN, COD, EUR, IND, KAZ, MNG, RUS, USA, ZAF |
| $CH_4$ | *Anthropogenic* | ARG, AUS, BRA, CHN, EUR, IDN, IND, IRN, MEX, PAK, RUS, USA |
| | *Fossil* | CHN, EUR, GULF, IDN, IND, IRN, KAZ&TKM, MEX, NGA, RUS, USA, VEN |
| | *Agriculture and Waste* | ARG, BGD, BRA, CHN, EUR, IDN, IND, MEX, PAK, RUS, THA, USA |
| $N_2O$ | *Anthropogenic* | AUS, BRA, CHN, COD, COL, EUR, IDN, IND, MEX, SDN, USA, VEN |

Earth System
Science
Data

# 3 Results for net land CO$_2$ fluxes

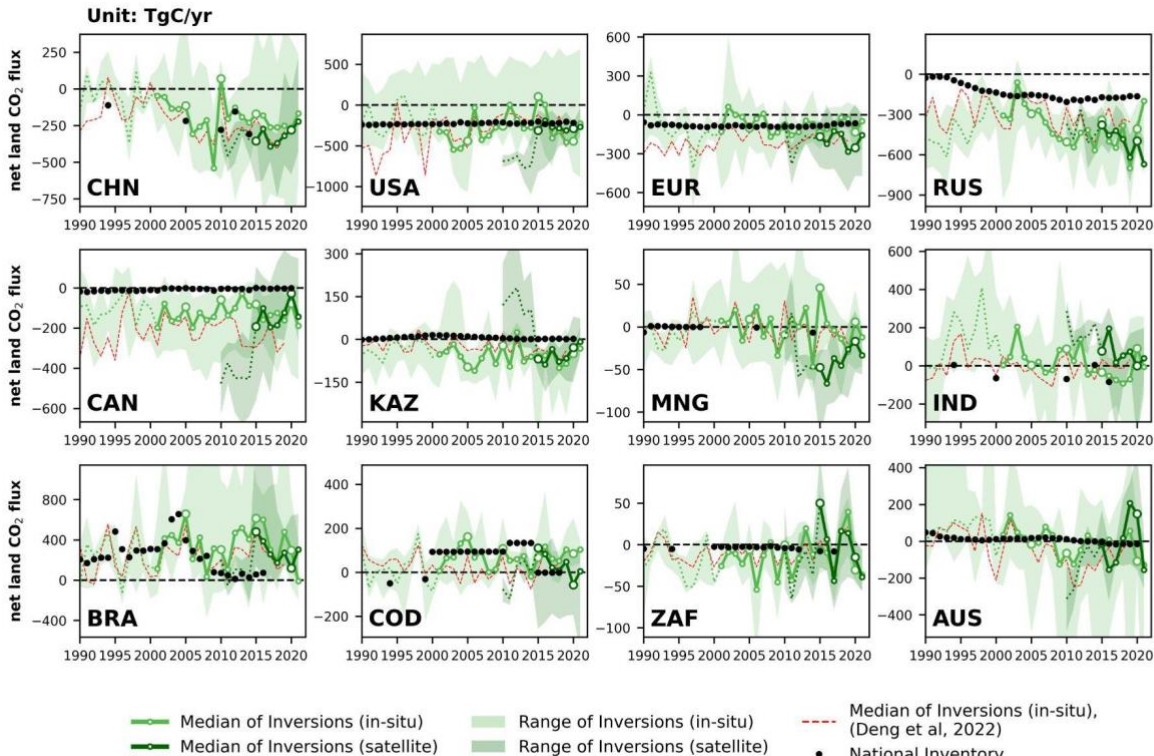


**Figure 3 | Net land CO$_2$ fluxes (unit: TgC yr$^{-1}$) during 1990-2021 from China (CHN), United States (USA), European Union (EUR), Russia (RUS), Canada (CAN), Kazakhstan (KAZ), Mongolia (MNG), India (IND), Brazil (BRA), Democratic Republic of the Congo (COD), South Africa (ZAF), and Australia (AUS).** By convention, CO$_2$ removals from the atmosphere are counted negatively, while CO$_2$ emissions are counted positively. The black dots denote the reported values from NGHGIs. The light green color denotes the in-situ-alone

CO$_2$ inversion (n=5) set while the dark green color denotes the set that uses satellite data (n=4). The green lines denote the median of land fluxes over managed land of CO$_2$ inversions, after adjustment of CO$_2$ fluxes from lateral transport by rivers, crop, and wood trade. When all inverse models within the inversion sets (in-situ: n=5; satellite: n=4) have available data for the same time interval, their median values are depicted as solid green lines. Otherwise, when the inversion sets have incomplete inverse models within the time interval (in-situ: n<5; satellite: n<4), their median values are represented as dashed green lines. The shading area denotes the min-max range of inversions. The

red dashed lines denote the median of inversions presented by the previous study (Deng et al., 2022).

**Fig 3** presents the time series of land-to-atmosphere CO$_2$ fluxes for the selected countries listed in **Table 2**. The median of inversions across the 12 countries shows significant interannual variability, reflecting the impact of climate variability on terrestrial carbon fluxes and annual variations of land-use emissions.

The adjustments of lateral CO$_2$ flux generally tend to lower land carbon sinks or increase land carbon emissions, especially in

CHN, USA, EUR, RUS, CAN, IND, and BRA. However, even with these adjustments, in countries of temperate latitudes, the median values of the five in-situ-alone inversion ensemble all indicate a net carbon sink during the 2010s, such as CHN with



a median sink of 180 ± 100 TgC/yr, USA (210 ± 180 TgC/yr), EUR (90 ± 50 TgC/yr), RUS (490 ± 100 TgC/yr) and CAN (110 ± 40 TgC/yr). In CHN, despite only 5 reported values to UNFCCC, NGHGIs show a good agreement with the inversion results, with both NGHGIs and inversions exhibiting an overall increase in carbon sink over the study period. However, during

2015-2021, the median values of the satellite-based inversion ensemble show a higher carbon sink of 320 ± 60 TgC/yr than those from in-situ inversion results (220 ± 50 TgC/yr) in CHN. In IND, there are also only five reported estimates from the NGHGIs. The in-situ inversion results indicate that India exhibited fluctuations between being a carbon source and a carbon sink during the period of 2001-2014 (40 ± 70 TgC/yr). During 2015-2019, the in-situ inversion results in IND show a median carbon sink of 65 ± 20 TgC/yr, however, the median reverted to being a carbon source of 91 TgC/yr (ranging from a sink of

350 to a source of 260) in 2020. In contrast, the median values of satellite-based inversion ensemble indicate a carbon source of 65 ± 60 TgC/yr during 2015-2021 in IND.

As Annex I countries, USA, EUR, RUS, CAN, and KAZ have continuously reported annual NGHGIs since 1990. The NGHGIs reported values for the USA and CAN indicate a decline trend (Mann-Kendall Z=-0.6, p<0.01) of carbon sinks by an annual average rate of 0.7 TgC/yr$^2$ and 0.5 TgC/yr$^2$. Like in Deng et al. 2022, we found that the carbon sink of Canada's managed

land is significantly larger (-125 ± 45 TgC/yr over 2001-2021 from in-situ inversions)  that the NGHGIs reports (5 ± 4 TgC/yr over 2001-2021). Part of this difference could be due to the fact that Canada decides in its inventory not to report fire emissions as they are considered to have a natural cause. Doing so, Canada also excludes recovery sinks after burning and those recovery sinks could surpass on average fire emissions, although remote sensing estimates of post fire biomass changes suggest

that fire emissions have exceeded regrowth on average in Western Canada and Alaska until ≈ 2010 (Wang et al.,

2021). One reason for the difference may be that the NGHGI used old growth curves for forests, potentially underestimating the actual forest growth; Another reason for the difference may be shrubland and natural peatland carbon uptake and possibly an underestimated increase of soil carbon in the national inventory. For the USA we have a good agreement between inversions (-290 ± 180 TgC/yr for in-situ over 2001-2021) and the NGHGIs data (-220 ± 10 TgC/yr over 2001-2021) with the inversion showing much more interannual variability, the US being a net source of carbon in the years 2011, 2015 and 2016 from the

median of in-situ inversons. The lower variability in the NGHGIs data reflects the 5-years averaging of C stock changes by the national forest inventory. In EUR, the new in-situ inversion ensemble gives a lower carbon sink than the previous one (red line in **Fig 3**, see discussion in section 6.1), now being in good agreement (-75 ± 60 TgC/yr) with NGHGIs (-85 ± 10 TgC/yr) over 2001-2021. The OCO2 satellite inversions give a higher sink than in-situ inversions by -200 ± 85 Tg C/yr, possibly because the in-situ surface network does not cover Eastern European countries which have a larger NEE than Western

European ones, whereas OCO2 data have  a more even coverage of the continent, as discussed by Winkler et al. (2023) ( see their Fig. 2 showing that OCO2 inversions have a similar NEE than in-situ ones in Western Europe but a larger mean NEE uptake in Eastern Europe).

In contrast, the NGHGIs in RUS reports a rapid trend of increasing sink by a rate of 4.6 TgC/yr$^2$ (Mann-Kendall Z=0.69, p<0.01) during 1990-2020, supported by the significant strong correlation with the medians of in-situ inversion ensemble





(ρ=0.7, p<0.01) during 2001-2020. However, the median values for both the in-situ (480 ± 100 TgC/yr) and satellite-based (450 ± 90 TgC/yr) inversion ensemble over RUS indicate larger larger land carbon sinks than those reported in the NGHGIs (178 ± 11 TgC/yr) during 2011-2020. For KAZ, the NGHGIs suggest that managed land is a slight carbon source (6 ± 5 TgC/yr) during 2000-2020. However, the median values for both satellite-based and in-situ inversion ensemble indicate a carbon sink of 53 ± 29 TgC/yr and 57 ± 33 TgC/yr, respectively, during 2015-2021 and 2001-2021. It is worth noting that the

satellite-based inversion results for USA, CAN, and KAZ all exhibit shifts in their fluxes between 2010 and 2015 compared to the results after 2015. This is attributed to the use of different satellite data and the number of different ensembles during these periods. Before 2015, only GOSAT was available, and only 2 out of 4 systems were available. After the OCO-2 record started, in September 2014, the satellite-driven inversion set only assimilated OCO-2. This indicates that inversion results based on GOSAT data are not consistent at the country scale with OCO-2 inversions. As a result, we can compare OCO-2

inversions with NGHGIs since 2015, but not the trends from inversions using GOSAT and/or OCO-2 inversions since 2009. In BRA, both the NGHGIs reports (239 ± 166 TgC/yr during 1990-2016) and inversion results (in-situ: 350 ± 190 TgC/yr during 2001-2021; satellite-based: 280 ± 120 TgC/yr during 2015-2021) indicate that the country has been a net carbon source since 1990. The carbon source from managed land in Brazil increased from the late 1990s, reaching a peak around 2005 according to NGHGIs (677 TgC/yr). This evolution is confirmed by in-situ inversions with a source peaking in 2005 (~650

TgC/yr). The net carbon source from inversions then decreased from 2005 to 2011, which is consistent with the observed reduction in deforestation due to forest protection policies implemented by the Brazilian government. This is an encouraging result as the inversions did not explicitly consider land use emissions in their prior assumptions, although some included an estimate of carbon released by fires in their prior which is part of land-use emissions in Brazil. Since NEE is defined as all land fluxes except fossil fuel emissions, NEE from all inversions nevertheless include land use emissions from deforestation,

degradation emissions and fire emissions including fires from deforestation, degradation and other fires. After 2011, inversions show a new increase in land emissions, with a peak during the 2015-2016 El Niño drought. There have been higher average land emissions thereafter. These ongoing changes may be attributed to various factors such as the legacy effects of drought leading to increased tree mortality (Aragão et al., 2018), higher wildfire emissions (Naus et al., 2022; Gatti et al., 2023), carbon losses from forest degradation, and climate change-induced reductions in forest growth due to regional drying and warming in

the southern and eastern parts of the Amazon (Gatti et al., 2021). From 2011 to 2016, the NGHGIs reports indicate that carbon emissions from Brazilian managed lands were stable at around 47 TgC/yr. However, the medians of in-situ inversions suggest that carbon emissions rapidly increased from ~100 TgC/yr in 2011 to ~600 TgC/yr in 2016, which peaked in 2015 (~610 TgC/yr). From 2016 to 2021, the medians for both in-situ and satellite inversion results show a decrease in carbon emissions from 2016 to 2018 but a transient peak in 2019, a year with large fires (Gatti et al., 2023) (in-situ: 480 TgC/yr; satellite: 270

TgC/yr). Then carbon emissions decreased again until 2021, which experienced wetter conditions and fewer fires (Peng et al., 2022); The in-situ inversion results show a continuous decrease to -10 TgC/yr in 2021, while the satellite inversion results showed a persistent source carbon anomaly of 300 TgC/yr. We emphasize moreover that available $CO_2$ observations from a network of aircraft vertical sampling (Gatti et al., 2021) were not used to constrain the inverse models used here.





For COD, the available NGHGIs data indicates that before 2000, the country's managed lands were a net carbon sink (50
TgC/yr in 1994 and 30 TgC/yr in 1999). Since 2000, the NGHGIs reports indicated three stages of different levels of $CO_2$ flux,
which COD managed land was a carbon source during 2000-2010 ($95 \pm 0.5$ TgC/yr), a larger carbon source during 2011-2014
($135 \pm 0.1$ TgC/yr), and a very small sink during 2015-2018 ($-1.2 \pm 0.1$ TgC/yr). The medians of in-situ inversion ensemble
indicate a similar annual average carbon source ($70 \pm 45$ TgC/yr) during 2001-2021 with the NGHGIs, despite the few
observations over Africa (Byrne et al., 2023). In the recent decade, satellite inversion results from 2015 to 2021 indicate a
smaller source ($30 \pm 55$ TgC/yr) compared to the in-situ results ($85 \pm 25$ TgC/yr). Moreover, the satellite inversion results
indicate a sink anomaly in 2020 (-60 TgC/yr) which is not found in the in-situ inversions. The sink anomaly in 2020 from the
satellite inversions is consistent with wetter conditions during that year over COD.

For ZAF, the NGHGIs show a stable very small sink of 3 TgC/yr during 1990-2010 that doubled from 4 TgC/yr in 2010 to 8
TgC/yr in 2017, while the in-situ inversion results indicate large fluctuations from a carbon sink (especially peaked in 2006,
2009, 2011, 2017 and 2021) to a small carbon source (e.g., in 2013, and 2018-2019). From 2015 to 2021, the satellite-based
inversion results are consistent with the in-situ results for annual variability ($\rho$=0.8, p<0.05), which is a good sign of the
consistency between different atmospheric observing systems. During the transition to El Niño conditions and drought from
2014 to 2015, however, the satellite-based inversion results indicate a switch from a carbon sink to a source anomaly of 50
TgC/yr in ZAF which is not seen in the in-situ inversions.

In AUS, the NGHGIs data shows a land source of carbon from 1990 to 2012, which decreased over time (from 48 TgC/yr in
1990 to 1 TgC/yr in 2012) and changed into a carbon sink since 2013 (that increased from a sink of 1 TgC/yr in 2013 to 15
TgC/yr in 2020). However, the in-situ inversions indicate fluctuations between a carbon source and a sink with an annual
average small sink of $10 \pm 71$ TgC/yr observed over the period of 2001-2021, except for 2009-2011, the medians of in-situ
inversions reveal a strong carbon sink of $105 \pm 35$ TgC/yr. Between 2010 and the strong La Niña year of 2011, the medians of
515 in-situ inversion ensemble from the previous study (Deng et al., 2022) showed an increase in carbon uptake of 145%. This
high carbon sink persisted in 2012, which was a dryer year with maximum bushfire activity. However, in this study, the
medians of updated in-situ inversion ensemble indicate that there is a sink anomaly in 2011 followed by a source anomaly in
2013, which appears to be more realistic. 2019 was the driest and hottest year recorded in Australia, including extreme fires at
the end of 2019 (Byrne et al., 2021). As a result, the medians for both in-situ  and satellite inversion ensemble show a carbon
source anomaly in 2019, with 55 TgC/yr (ranging from a sink of 1060 to a source of 480) and 200 TgC/yr (raging from a sink
of 120 to a source of 320) respectively. When it comes to the wet La Niña year of 2021, the medians for both in-situ and
satellite inversion ensemble indicate that AUS managed land became a carbon sink of 130 TgC/yr (ranging from a sink of
1120 to a source of 25) and 150 TgC/yr (ranging from a sink of 260 to a source of 40).

Last, we give the global comparison between NGHGIs and inversions, using NGHGIs data compiled for all countries by Grassi
et al. (2023) which include Annex I countries reports, non-Annex I NC, BUR and NDCs.  The river correction is the only one
that changes the global NEE, because the global mean of $CO_2$ fluxes from wood and crop products is close to zero. The river-
induced $CO_2$ uptake over land that is removed from inversion NEE is equal to the C flux transported to the ocean at river



mouths (0.9 GtC/yr in our estimate, close to the value of Regnier et al. 2022).The (in-situ) inversions without the river correction give a global NEE sink of 1.8 GtC/yr over 2001-2020, managed land: 1.3 GtC/yr (72% of total), unmanaged land:

0.5 GtC/yr (28%). The in-situ inversions with the river correction study give a global NEE sink of 0.91 GtC/yr, managed land:0.51 GtC/yr (56% of total), unmanaged land 0.4 GtC/yr (44% of the total)  This is an important update from Deng et al. 2022 where the river CO2 flux correction was not applied separately to managed / unmanaged lands. Because managed lands have a much larger area than unmanaged ones and because of the spatial patterns of the CO2 sinks in the river correction are distributed with MODIS NPP which has low values in unmanaged lands of northern Canada and Russia, the river correction

reduces strongly the C storage change with respect to NEE over managed lands, and marginally in unmanaged lands.. Inventory data recently compiled by Grassi et al. (2023) indicates a similar global land sink (on managed land) of 0.53 GtC yr$^{-1}$ with gap-filled data during the same period than the inversions with our improved river correction.

## 4 Results for anthropogenic CH$_4$ emissions

### 4.1 Total anthropogenic CH$_4$ emissions

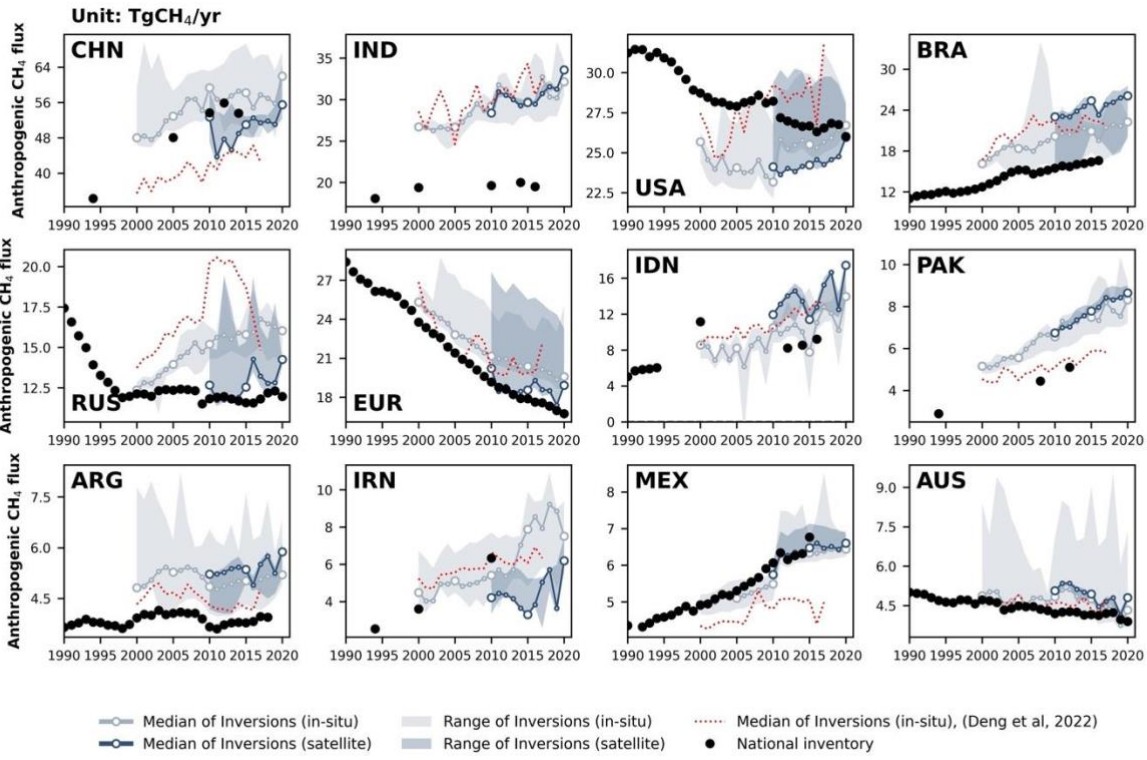


**Figure 4. Total anthropogenic CH$_4$ fluxes for the 12 top emitters: China (CHN), India (IND), United States (USA), Brazil (BRA), Russia (RUS), European Union (EUR), Indonesia (IDN), Pakistan (PAK), Argentina (ARG), Iran (IRN), Mexico (MEX), and**



**Australia (AUS).** The black dots denote the reported values from NGHGIs. The light and dark blue lines/areas denote the median and maximum-minimum ranges of in-situ and satellite-based $CH_4$ inversions based on EDGARv6.0 as the prior respectively.

**Fig 4** presents the variations in anthropogenic $CH_4$ emissions for the 12 selected countries, where these emissions are summing the sectors of agriculture and waste, fossil fuels, and biofuel burning. The distribution of emissions is highly skewed even among the top 12 emitters, with the largest and most populated countries such as CHN, IND, USA, BRA, RUS, and EUR which emits more than 10 $TgCH_4$/yr annualy, while other countries have smaller emissions (ranging from 3 to 10 $CH_4$/yr) that are more challenging to quantify through inversions. During 2010-2020, CHN has the highest total anthropogenic emissions

at around $50 \pm 3.5$ Tg $CH_4$/yr, followed by IND with $30 \pm 1.4$ Tg $CH_4$/yr, USA with $24 \pm 0.6$ Tg $CH_4$/yr, BRA with $24 \pm 1.2$ Tg $CH_4$/yr, EUR with $19 \pm 0.7$ Tg $CH_4$/yr, IDN with $14 \pm 0.9$ Tg $CH_4$/yr and RUS with $13 \pm 0.9$ Tg $CH_4$/yr, according to the medians of satellite-based inversion ensemble based on EDGARv6.0 as prior. The remaining countries have emissions of approximately 5 Tg $CH_4$/yr. In general, the difference between NGHGIs and inversions aligns in the same direction based on both satellite and in-situ inversions. This provides some confidence for using inversions to evaluate NGHGIs as the satellite

observations are independent from in situ networks. Overall, satellite-based inversions may be more robust across most countries due to better observation coverage, except in EUR and the USA where the in-situ network is more extensive.

Developing countries, such as CHN, IND, BRA, IDN, PAK, IRN and MEX, show a rapid increase in anthropogenic $CH_4$ emissions supported by reported values from NGHGIs and results from inversions. In CHN, the reported values from NGHGIs (when available) generally align with the results obtained through inversions (e.g., during 2010-2015, NGHGIs: $54 \pm 1.3$ Tg

$CH_4$/yr, in-situ: $58 \pm 1.2$ Tg $CH_4$/yr, satellite-based: $48 \pm 3.4$ Tg $CH_4$/yr). During 2010-2020, the median values for the in-situ and satellite-based inversion ensemble show a similar increase trend at an annual growth rate of 0.28 Tg $CH_4$/yr$^2$ and 0.26 Tg $CH_4$/yr$^2$ respectively, although the medians of in-situ inversion ensemble ($58 \pm 2.0$ $TgCH_4$/yr) were slight higher than the satellite-based ensemble ($50 \pm 3.5$ $TgCH_4$/yr). However, in 2020, the medians of the emission estimates for both in-situ and satellite-based inversions reveal a rapid increase by 9% and 11% compared to 2019 in CHN, indicating a possible surge in

anthropogenic methane emissions for that year, possibly an artifact from the fact that the decreased OH sink in 2020 is not well acoounted for here. Indeed OH IAV were not prescribed to all inversions, and when accounted for the OH IAV prescribed (based on Patra et al., 2021) was much smaller than those suggested by recent studies (e.g., Peng et al., 2022). As a result overestimating the sink in the inversions leads to overestimated surface emissions. The surge in emissions could also be due to spin-down, the last six month to one year of inversions being less constrained by the observations, even though the inversion

period covered up to June 2021.

In IND, PAK and MEX, there is good agreement (r>0.8, p<0.01) between the in-situ and satellite-based inversion ensembles (respectively, $31 \pm 1.2$ Tg $CH_4$/yr and $30 \pm 1.4$ Tg $CH_4$/yr in IND, $8 \pm 0.7$ Tg $CH_4$/yr and $7 \pm 0.5$ Tg $CH_4$/yr in PAK, and $6 \pm 0.2$ Tg $CH_4$/yr and $6 \pm 0.3$ Tg $CH_4$/yr in MEX), while both of them present a significant increasing trend of anthropogenic methane emissions in these countries (Mann-Kendall p<0.05). However, when comparing to NGHGIs values, the inversion

results in IND and PAK indicate >50% larger emissions than the values reported from the NGHGIs during 2010-2020. In contrast, values reported from the NGHGIs ($6 \pm 0.2$ Tg $CH_4$/yr) by MEX also show good agreement with the inversion results.



In BRA, IDN and ARG, the medians for in-situ and satellite-based inversion ensembles show good consistency (r=0.8, p<0.01) in these two countries, while satellite-based inversion results are generally higher than the in-situ inversion results. Specifically, in BRA, the satellite-based inversions (24 ± 1.2 Tg $CH_4$/yr) were 16% higher than the in-situ inversions (21 ± 0.8 Tg $CH_4$/yr)

and 52% higher than the NGHGIs estimation (17 ± 0.4 Tg $CH_4$/yr) during 2010-2020, possibly owing to difficulties for inversions to separate between natural (wetlands, inland waters) and anthropogenic sources in this country, and possible flaws in the prior used for natural and anthropogenic fluxes. In IDN, NGHGIs reported a significant continuous upward trend at an annual average growth of 0.3 Tg$CH_4$/yr, with a noticeable positive outlier in 2000. The medians for both in-situ and satellite-based inversion ensembles also indicate an upward trend in IDN, but both of them present sudden dips in anthropogenic

methane emissions in 2015 and 2019 by 15~23% and 16~25%, compared to the previous year respectively. It is unlikely that anthropogenic activities could contribute such large year to year variations except for different flooded areas used for rice paddies. In ARG, the satellite-based inversion results also indicate two sudden dips in 2016 and 2019, however, such pattern was not found in the in-situ inversion results. A cause of year to year variations from inversions is the lack of in-situ sites and variable cloud cover affecting the density of GOSAT data.

Regarding IRN, NGHGIs only provided data for three years (1994, 2000, and 2010), making it difficult to compare with inversion results. However, NGHGIs show a rapid growth in anthropogenic $CH_4$ emissions (+9.4%/yr) during this period. There are significant differences between inversion results and for IRN, with satellite inversions generally giving lower emissions than in-situ inversions and different trends. Satellite inversions suggest a declining trend between 2010 and 2015, followed by a fluctuating increase until 2020. In contrast, in-situ-based inversions ( by any nearby measurement stations, thus

likely reflecting the prior trend) show a rapid rise in emissions after 2010, reaching a peak in 2018, followed by a decline. NGHGIs for RUS indicate that anthropogenic $CH_4$ emissions have been reduced during the 1990s and remained stable since 2000 (12.0 ± 0.3 Tg $CH_4$/yr during 2000-2020), which is similar with the trend observed from satellite-based inversion results (12.7 ± 0.9 Tg $CH_4$/yr during 2000-2020). However, in 2016, there was a sudden increase of emissions in satellite inversion results (+14% increase from 12.5 in 2015 to 14.2 Tg $CH_4$/yr in 2016), followed by a gradual decline, and then a new increase

in 2020 (+11% increase from 12.8 Tg $CH_4$/yr in 2019 to 14.3 Tg $CH_4$/yr in 2020). This recent change was not observed in the in-situ inversion results or the NGHGIs.

For USA, AUS, and EUR, NGHGIs reported a slow declining trend (EUR: 0.4 Tg $CH_4$/yr; USA: 0.2 Tg $CH_4$/yr; AUS: -0.04 Tg $CH_4$/yr) in anthropogenic $CH_4$ emissions. In the case of the USA, inversion-derived emissions are slightly lower than NGHGIs (in-situ-based: 9.3% lower during 2000-2020; satellite-based: 11.4% lower during 2010-2020). However, both

ground-based and satellite-based inversions indicate that anthropogenic $CH_4$ emissions have remained relatively steady since 2000, without reflecting the slow decline reported by NGHGIs. In EUR, NGHGIs indicate that anthropogenic $CH_4$ emissions have been decreasing rapidly since 1990 (-1.4%/yr), consistent with the trend obtained from inversion results. However, in-situ inversion emissions are on average slightly higher than NGHGIs, and this difference has been gradually increasing from 7.7% in the 2000s to 14.5% in the 2010s.

**4.2 Fossil CH₄ emissions**

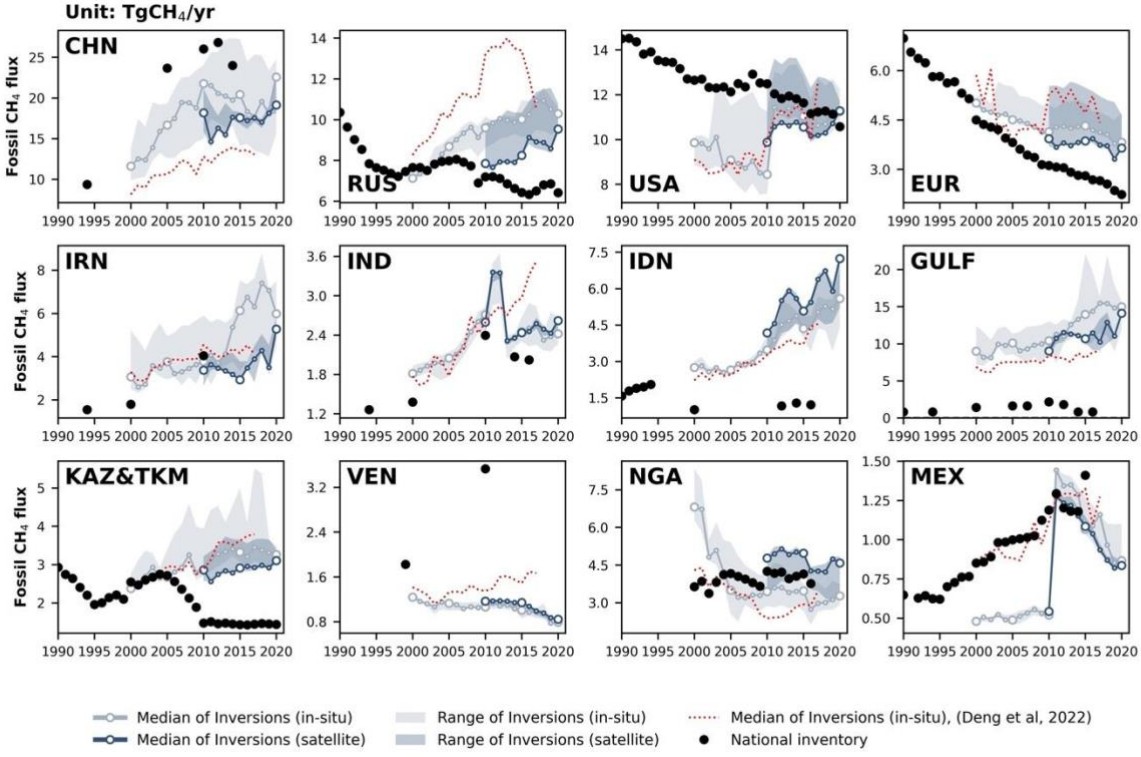

**Figure 5. CH₄ emissions from the fossil fuel sector from the top 12 emitters of this sector: China (CHN), Russia (RUS), United States (USA), European Union (EUR), Iran (IRN), India (IND), Indonesia (IDN), Persian Gulf countries (GULF = Saudi Arabia + Iraq + Kuwait + Oman + United Arab Emirates + Bahrain + Qatar), Kazakhstan & Turkmenistan (KAZ&TKM), Venezuela (VEN),**

**Nigeria (NGA), and Mexico (MEX).** The black dots denote the reported value from the NGHGIs. In the NGHGI data shown in Fig 5 for GULF, Saudi Arabia reported four NGHGIs in 1990, 2000, 2010, and 2012, Iraq reported one in 1997, Kuwait reported three in 1994, 2000, and 2016, Oman reported one in 1994, United Arab Emirates reported four in 1994, 2000, 2005 and 2014, Bahrain reported three in 1994, 2000 and 2006, and Qatar reported one in 2007. The reported values are interpolated over the study period to be summed up and plotted in the figure. For KAZ&TKM, the reported values of Turkmenistan during 2001-2003, 2005-2009, 2011-2020 are interpolated and added to

annual reports from Kazakhstan, an Annex I country for which annual data are available. Other lines, colors and symbols as **Fig 4**.

**Fig 5** presents the fossil CH₄ emissions for the top 12 emitters from the fossil sector based on EDGARv6.0 as the prior. The largest emitter is CHN, mainly from the sub-sector of coal extraction, followed by RUS and USA. In CHN, the in-situ (20 ± 1.6 Tg CH₄/yr) and satellite inversions (17 ± 1.3 Tg CH₄/yr) emissions in the 2010s are 24% and 35% lower than in the NGHGIs (26 ± 1.5 Tg CH₄/yr), respectively. The NGHGIs in CHN suggest a decrease from 28 in 2012 to 24 TgCH₄/yr in

2014. However, both in-situ and satellite inversion results indicate an increasing trend since 2018. In IND and IDN, NGHGIs report a decreasing trend during the study period, while inversions suggest a rapid increase in IDN and a stable value in IND after a peak in 2012. In IND, satellite inversions suggest a peak of fossil CH₄ emissions during 2011-2012, which then dropped





in 2013 and remained stable afterward. In IDN, both in-situ and satellite inversions indicate a fluctuating trend, with a significant drop between 2015 and 2019. In RUS, both in-situ and satellite inversion-based estimates of fossil fuel emissions are higher than NGHGIs, and show an increasing trend, while NGHGIs report a decreasing trend. This discrepancy may be due to inversion problems for separating between wetland emissions and gas extraction industries both located in the Yamal peninsula area, or leaks not captured in NGHGIs. In USA, NGHGIs overall show a significant declining trend (Mann-Kendall Z=-0.8, p<0.01). In-situ inversion estimates of fossil fuel emissions are 26% lower than NGHGIs during 2000-2010, and remained consistent until around 2011. Nearly all in-situ inversions show a jump in fossil fuel emissions in 2011. In EUR, both NGHGIs and inversion results demonstrate a consistent declining trend. However, starting from 2010, both in-situ and satellite inversions are higher than NGHGIs reports.

Major oil-producing countries in the persian Gulf are too small compared to the model resolution to be studied individually. Hence, NGHGIs from the GULF countries (Saudi Arabia, Iraq, Kuwait, Oman, United Arab Emirates, Bahrain, and Qatar) were grouped and show much lower emissions compared to inversion results. In the 2010s, in-situ and satellite inversions estimate that emissions in GULF were 9 times and 8 times higher than the estimates reported in NGHGIs, respectively. This huge under-reporting of emissions in GULF could be partly attributed to the omission of ultra-emitters in NGHGIs. Indeed, recent studies by Lauvaux et al. (2022) have identified more ultra-emitters and larger emission budgets from ultra-emitters in Qatar, Kuwait, and Iraq. In KAZ&TKM, grouped together because of their rather small individual areas, both in-situ ($3.3 \pm 0.2$ Tg $CH_4$/yr) and satellite ($2.9 \pm 0.1$ Tg $CH_4$/yr) inversions estimate emissions to be 2 times higher than NGHGIs ($1.5 \pm 0.1$ Tg $CH_4$/yr) in the 2010s. Similarly, KAZ is located downwind of TKM, which has a high share of ultra-emitters. The global inversions operating at a coarse resolution may misallocate emissions from TKM to KAZ. It is worth noting that KAZ has two in-situ stations for $CH_4$ measurements, whereas the GULF countries lack in-situ station networks. On the other hand, the GOSAT satellite provides a dense sampling of atmospheric column $CH_4$ in the Persian Gulf region due to frequent cloud-free conditions. Therefore, GOSAT inversions can be considered more accurate than in-situ inversions for IRN, GULF countries, and KAZ&TKM. Additionally, it is important to note that GOSAT inversions generally give lower emissions than in-situ inversions in those countries. VENis a rare case where NGHGIs report much higher $CH_4$ emissions than inversions. While the uncertainty of GOSAT inversions (model spread) has decreased compared to the results reported by Deng et al. 2022, the gap between inversions and NGHGIs has increased . In 2010, NGHGIs reports of fossil $CH_4$ emissions in VEN were 298% higher than GOSAT inversions and 326% than in-situ inversions. We do not have a clear explanation for this large difference, except that VEN has strongly decreased oil and gas extraction due to sanctions curbing its crude production from 2.65 mb/d in 2015 to 0.57 mb/d in 2020 (OPEC, 2023), which may not be reflected in their NGHGIs. In NGA and MEX, NGHGIs estimates fall between the median of in-situ and satellite inversions during 2010-2020. However, in MEX, the in-situ inversion was 50% lower than NGHGIs in the 2000s and showed a sudden large increase in 2010.





## 4.3 Agriculture and waste CH₄ emissions

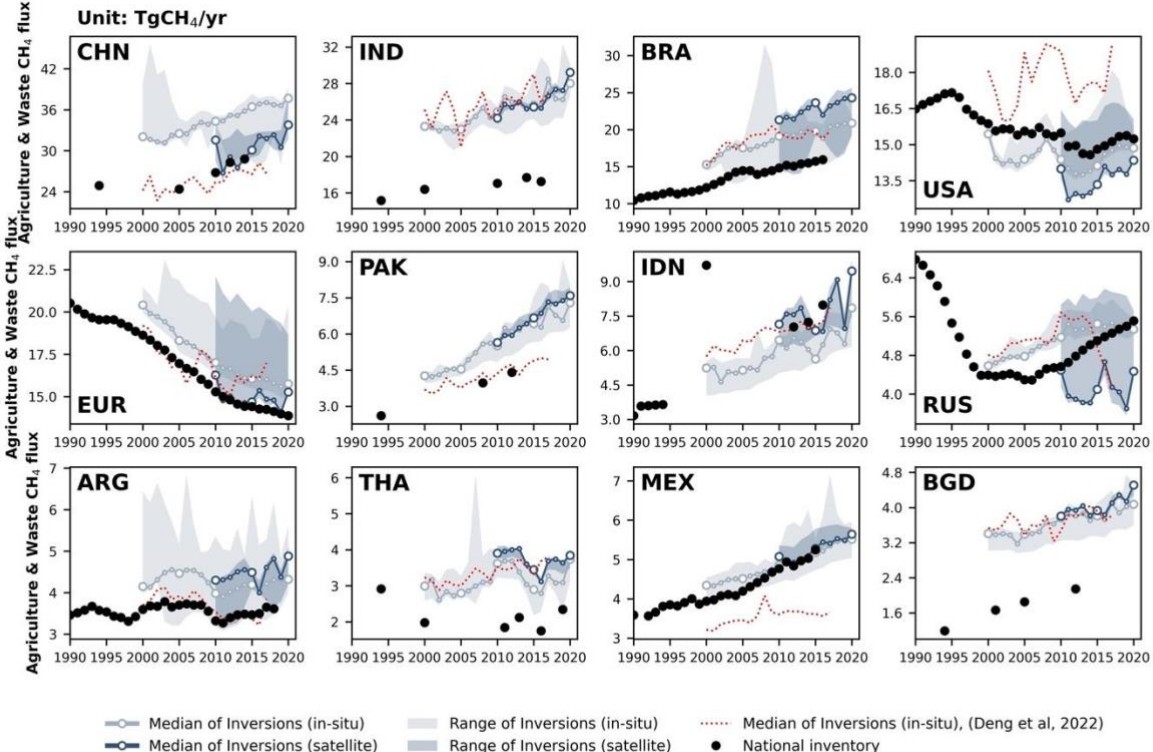

**Figure 6. CH₄ emissions from agriculture and waste for the 12 largest emitters in this sector, China (CHN), India (IND), Brazil (BRA), United States (USA), European Union (EUR), Pakistan (PAK), Indonesia (IDN), Russia (RUS), Argentina (ARG), Thailand (THA), Mexico (MEX), and Bangladesh (BGD).** The black dots denote the reported estimates from NGHGIs. Other lines, colors, and symbols as **Fig 4**.

**Fig 6** presents CH₄ emissions of the Agriculture and aste sector for the top 12 emitters of this sector. In all countries except for the USA and RUS, the values reported by NGHGIs are systematically lower than the inversion results. The results from the previous ensemble of in-situ inversions (red dotted line) are consistent with those of the inversions used in this study except in the USA where previous inversions are 3.2 TgCH₄/yr higher, in RUS where they show a drop after 2015 although they remain in the range from the new satellite and in-situ inversions, and in MEX where they are systematically lower by 1.6 TgCH₄/yr.

In CHN, the most recent NGHGIs reports in 2012 and 2014 estimate agriculture and waste emissions at 28 Tg CH₄/yr, which is close to satellite inversions (28 ± 1 TgCH₄/yr) but 22.4% lower than the median in-situ inversions (35 ± 0.5 TgCH₄/yr) and closer to their minimum value. The trend in agricultural and waste emissions is consistent between inversions and NGHGIs for CHN. In IND, inversions consistently show higher emissions than NGHGIs by approximately 50% and indicate an increasing trend during 2000-2020, whereas the NGHGI last communication being for 2016, it does not allow us to give a



recent trend. According to the national inventory of IND, enteric fermentation is the primary source of $CH_4$ emissions in the agriculture and waste sector, contributing 61% of emissions, with rice cultivation accounting for 20% and waste contributing 16%. A similar pattern is observed in BGD, where agricultural emissions are dominated by rice production (48% in 2012) and enteric fermentation (42% in 2012). Satellite and in-situ inversions estimate emissions in BGD are nearly double than those

reported by NGHGIs during 2001 and 2012, the last communication. The significant discrepancies between inversions and NGHGIs in IND and BGD may be attributed to potential underestimation of livestock or waste $CH_4$ emissions by NGHGIs. NGHGIs utilized the Tier 1 method and associated emission factors from the 2006 IPCC Guidelines for National Greenhouse Gas Inventories (IPCC, 2006). However, a recent study (Chang et al., 2021) found that estimates using revised Tier 1 or Tier 2 methods from the 2019 Refinement to the 2006 IPCC Guidelines for National Greenhouse Gas Inventories (IPCC, 2019)

give livestock emissions 48%-60% and 42%-61% higher for IND and BGD by 2010, respectively, compared to Tier 1 IPCC (2006) methods, which would bring bottom up emissions closer to inversions. In BRA, both satellite and in-situ inversions consistently estimate larger emissions than the NGHGIs by 34% and 29%, respectively, and show a consistent increasing trend over their study periods. In the USA, the medians of satellite and in-situ inversions are slightly lower than those of NGHGIs, but they exhibit a similar trend throughout the study period. The trend of inversions is comparable to the one of the NGHGIs

in BRA during their period of overlap, although there is no NGHGIs communication later than 2016. In ARG, PAK and THA, the medians of in-situ inversions show good consistency with satellite inversion results. Nevertheless, in-situ inversion emissions in the 2010s are, on average, 47% higher in PAK, 20% higher in ARG, and 64% higher in THA compared to the NGHGIs reports. In EUR, emissions from agriculture and waste were reported to have significantly decreased over time in the NGHGI data, mainly from solid waste disposal (Petrescu et al., 2021), a trend that is captured by inversions and is close to the

one of the NGHGIs over the study period. In contrast, emissions from agriculture and waste in RUS are reported to have a positive trend after 2010 by the NGHGI, with in-situ inversions producing a consistent trend from 2000 to 2014 but a sharp decrease thereafter, while satellite inversions are producing stable emissions, albeit lower than the NGHGIs and in-situ inversions after 2010.



## 5 Results for anthropogenic N₂O emissions

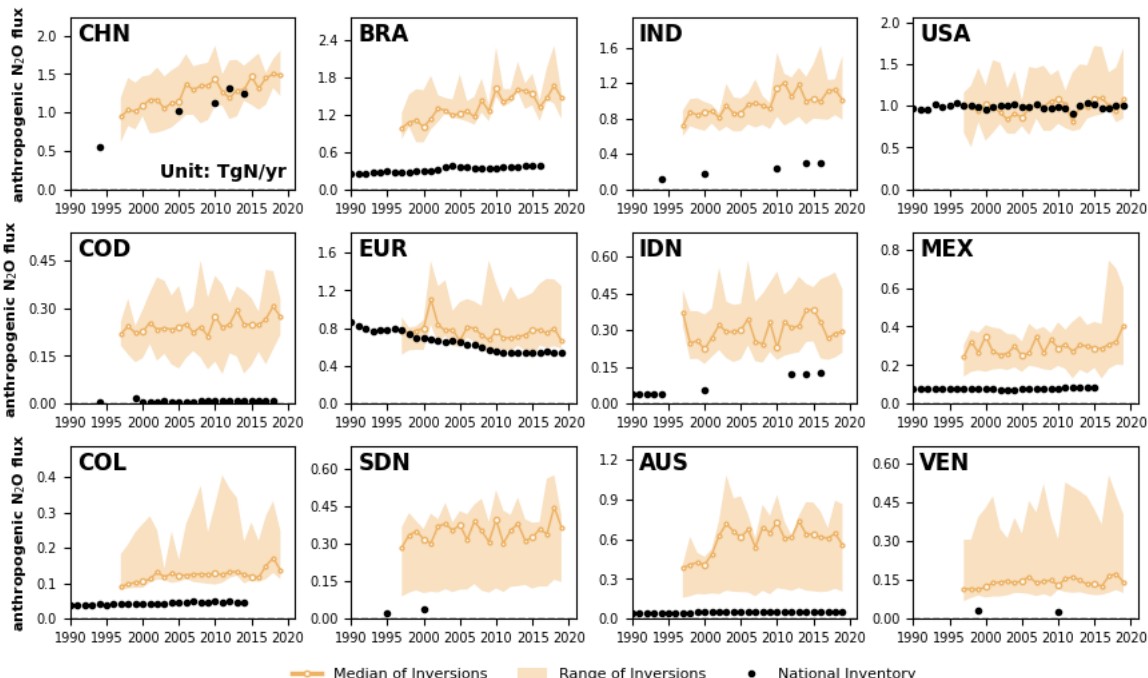


**Figure 7. Anthropogenic N₂O fluxes of the top 12 emitters: China (CHN), Brazil (BRA), India (IND), United States (USA), Democratic Republic of the Congo (COD), European Union (EUA), Indonesia (IDN), Mexico (MEX), Colombia (COL), Sudan (SDN), Australia (AUS), and Venezuela (VEN).** The black dots denote the anthropogenic emissions from the UNFCCC national greenhouse gas inventories. The thick orange lines and the light orange areas denote the median and the maximum-minimum ranges of

anthropogenic fluxes respectively among all N₂O inversions. We restricted our analysis to data starting from 1997 because it was the year when data from the all four inversion models are available.

We present the 12 countries/regions with the largest anthropogenic N₂O emissions in the world (**Fig 7**), which in total contribute approximately 55% of global anthropogenic N₂O emissions. The estimates from both NGHGIs and inversions in CHN, USA, and EUR demonstrate a relatively close match between NGHGIs and inversions (in-situ only). These three large

emitting countries/regions exhibit different trends in their anthropogenic N₂O emissions. In CHN, both NGHGIs and inversions indicate an increasing trend in anthropogenic N₂O emissions. In the USA, anthropogenic N₂O emissions seem to have reached a state of relative stability, with NGHGIs and inversion results showing similar mean values and lack of trends. In EUR, both NGHGIs and inversions show a declining trend in anthropogenic N₂O emissions, but from 2010 to 2020, the NGHGIs estimates are lower (20%) than the median values derived from inversion models, that is, the negative trend from inversions is less

pronounced than the one of NGHGIs. Most other selected countries display higher anthropogenic N₂O emissions from inversions than from NGHGIs (i.e., BRA, IND, COD, IDN, MEX, COL, SDN, VEN). These discrepancies in anthropogenic N₂O emissions are possibly attributable to factors that have been analyzed in our previous study (Deng et al., 2022). Firstly,



nearly all these non-Annex 1 countries utilize Tier 1 emission factors (EFs), which may underestimate emissions n when soil and climate dependence is taken into account (Cui et al., 2021). This has been noted in previous studies (Philibert et al., 2013; Shcherbak et al., 2014; Wang et al., 2020). Furthermore, the observed concave response of cropland soil emissions as a function of added N fertilizers may also contribute to underestimated emissions in NGHGIs, as the relationship is non-linear and higher than the linear relation used by NGHGIs in Tier 1 approaches (Zhou et al., 2015). In an improved reporting framework, EFs should also account for both natural and anthropogenic components, as they cannot be distinguished through field measurements, from which EFs are derived. However, in practice, EFs are mostly based on measurements made in temperate climates and soils from established croplands with few "background" emissions. Consequently, there could be a systematic underestimation of default IPCC EFs from tropical climates and for recently established agricultural lands, for which the IPCC EFs also have a huge uncertainty of up to $\pm75\%-100\%$. Another factor that might contribute to the discrepancy is the omission of emissions from reactive nitrogen contained in organic fertilizers (manure), for which NGHGIs do not provide specific details for non-Annex 1 reports. Lastly, anthropogenic indirect emissions (AIEs) from atmospheric nitrogen deposition and leaching of human-induced nitrogen additions to aquifers and inland waters are reported by Annex 1 countries using simple emission factors, but non-Annex 1 countries do not consistently report AIE. However, in AUS, the gap between inversions and NGHGIs is even expanded compared to our previous study. We do acknowledge that the density of the $N_2O$ in-situ network in tropical countries and around AUS is so low that inversions most likely are attracted to their priors. The use of a lower prior could thus also be consistent with scarce atmospheric observations, and we have only a low confidence on $N_2O$ inversion results for tropical countries and AUS.

# 6 Discussion

## 6.1 Comparing net land CO$_2$ flux estimates from different inversion model ensembles

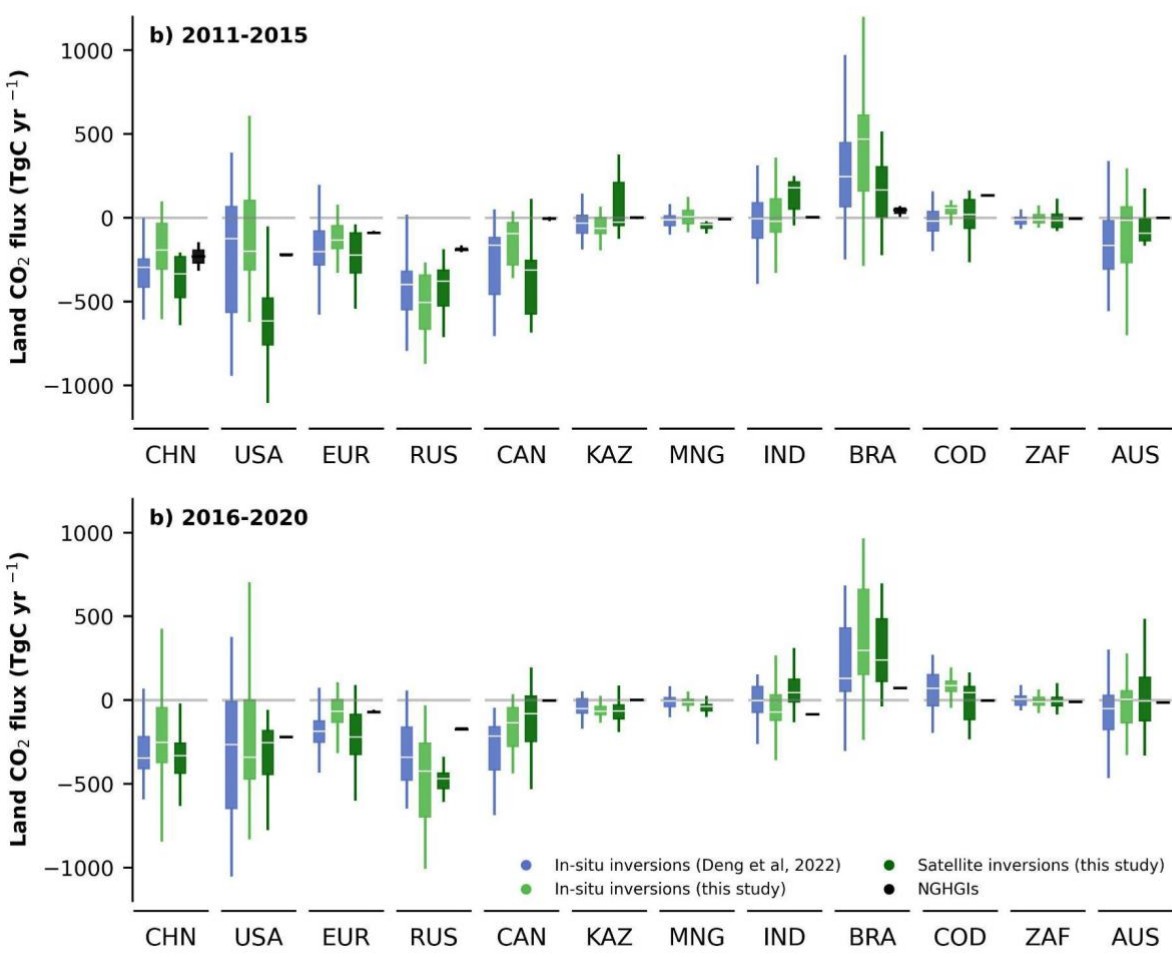

**Figure 8. Net CO$_2$ land fluxes during the period of a) 2011-2015; and b) 2016-2020 in China (CHN), United States (USA), European Union (EUR), Russia (RUS), Canada (CAN), Kazakhstan (KAZ), Mongolia (MNG), India (IND), Brazil (BRA), Democratic Republic of the Congo (COD), South Africa (ZAF), and Australia (AUS).** Blue boxes denote the in-situ inversion results from Deng et al. (2022) processed from Global Carbon Budget 2020 (Friedlingstein et al., 2020). Light green boxes denote the in-situ inversion results processed in this study, while dark green boxes denote the satellite inversion results. Black boxes denote the NGHGIs reported values. The white lines in



the boxes denote the medians of the land $CO_2$ fluxes. Note that the inversion results here have been adjusted by the lateral flux before the
comparison.

In this section, we compare four different estimates of land $CO_2$ fluxes during the period 2010-2020 (**Fig 8**), including: 1) medians of in-situ inversion results from our previous study (Deng et al., 2022), 2) medians of in-situ and 3) satellite-based inversion results processed in this study based on the Global Carbon Budget 2022 (Friedlingstein et al., 2022), and 4) NGHGIs. This enables a comparison of the median and range of our in-situ inversion results (n=5) with those from previous study (n=6),
and assesses the performance differences between satellite-based (n=4) and in-situ inversion models. To ensure a fair comparison and avoid anomalies in the satellite-based inversion results during 2010-2015 when some of these inversions used GOSAT after 2010 and then OCO-2 after 2015, we separate the analysis into two periods: 2011-2015 and 2016-2020.

The variations of yearly land $CO_2$ fluxes span a comparable range between the current and previous in-situ inversion ensembles, indicating that consistency of the inversion results, but the uncertainty within the new in-situ inversion ensemble
was not improved. However, examining the median values, results from the new in-situ inversion ensemble may be closer to NGHGIs in most countries (such as CHN, USA, EUR, CAN, KAZ, IND). This suggests that the new in-situ inversion ensemble used in this study has partially narrowed down the gaps between inversion results and NGHGIs compared to the previous one. However, in RUS and BRA, the difference between the median of in-situ inversion ensembles and NGHGIs has enlarged. For example, in RUS, median the new in-situ inversion ensemble indicate a larger carbon sink than those from Deng et al. (2022),
while the difference between median of in-situ inversions and NGHGIs increases 51% during 2011-2015 (from 208 TgC/yr to 314 TgC/yr) and 49% during 2016-2020 (from 168 TgC/yr to 249 TgC/yr). Conversely, in BRA, median of the new in-situ inversion ensemble indicate a larger carbon source, while the difference increases over 100% during 2011-2015 (from 200 TgC/yr to 423 TgC/yr) and nearly 300% during 2016-2020 (from 56 TgC/yr to 223 TgC/yr).

As for the inversion ensemble used in this study, in most countries, the variations of yearly land CO2 fluxes also span a similar
range between satellite-based inversion ensemble and in-situ inversion ensemble. However, in the cases of USA, RUS, CHN and BRA, the spread of satellite-based inversion results are narrower than those of in-situ inversion results, indicating a better consistency among available satellite-based inversion models, at least when similar satellite data are assimilated. In addition, in most cases, smaller difference were found between the median of inversion results and the NGHGIs. For countries with dense surface monitoring networks such as in the USA and EUR, the satellite-based inversion results show good agreement
in-situ inversion results. However, for countries with sparse station coverage like KAZ and MNG, satellite-based inversion results could provide more reliable estimates due to more extensive spatial sampling from satellites, although the medians of satellite-based inversion results indicate larger carbon sinks and larger differences compared with NGHGIs (than for in-situ inversion results). In USA and CAN, the difference during 2011-2015 (only GOSAT period) between in-situ and satellite-based inversion ensembles are larger than that during 2016-2020 (OCO2 period). This can be attributed to the use of different
satellite data during these periods and different numbers of ensemble members. Before 2015, only GOSAT was available, and only 2 out of 4 systems. The inversion of OCO-2 data starting in 2014 result in a better alignment among OCO-2 ACOS v10 inversions, indicating the in-situ and satellite evaluations were similar (Byrne et al., 2023).

Earth System
Science
Data

Open Access | Discussions

## 6.2 Adjustment of the national managed land masks to separate the net land CO₂ flux estimates

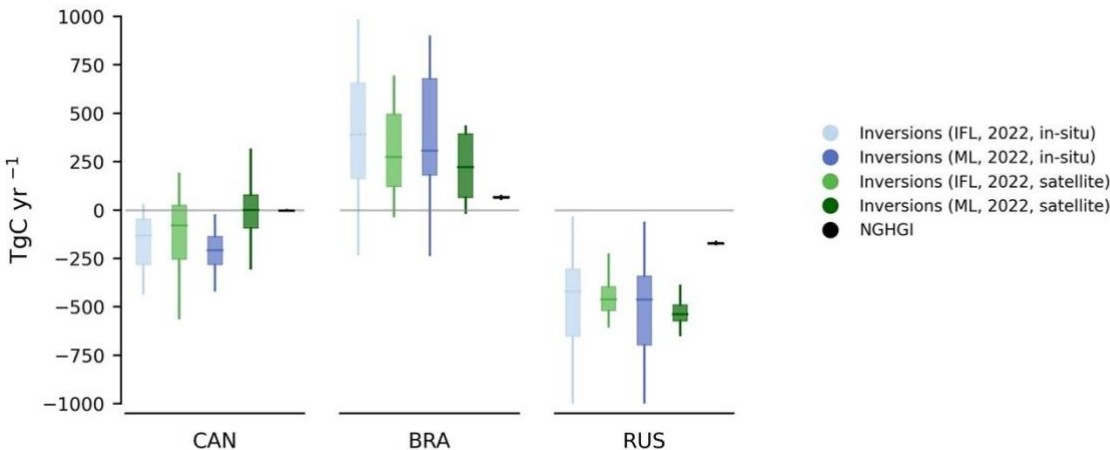

**Figure 9. Net CO₂ land fluxes during the period of 2015-2020 in Canada (CAN), Brazil (BRA), and Russia (RUS).** 'IFL' stands for using the intact forest landscape data as a mask for non-managed land to extract land CO₂ flux from managed land and 'ML' indicates the adjusted mask used by Grassi et al. (2023) to extract land CO₂ flux from managed land. The 'in-situ' stands for inversion results using in-situ observations, and 'satellite represents inversions using satellite observations. Note that the inversion results here have been adjusted by the lateral flux before the comparison.

Following the method proposed by Grassi et al. (2023), we updated in this study the managed land mask for CAN and BRA by using maps of managed land derived from NGHGI, and for RUS by adjusting tree-cover threshold in the tree cover map from Hansen et al. (2013) to match the average area of managed land per Oblast (province) that is used for the NGHGIs. Thus, the new mask is now more consistent with the definition of managed land in the NGHGIs for these three countries, so that can further analyze the impacts of different definitions of managed land masks to separate the managed land CO₂ fluxes

in inversions (**Fig 9**). Generally, in Russia (RUS) and Canada (CAN), the managed land CO2 fluxes extracted from the new mask are closer to NGHGIs than those separated by the previous mask used by Deng et al. 2022. In addition, in Brazil (BRA), adjusting the national managed land mask resulted in greater land carbon emissions, increasing the gap with NGHGIs. However, the improvement of the managed land mask in this study is still not able to explain all the existing discrepancy between inversion estimates and NGHGIs, in which the sources and reasons for these differences and uncertainties still need

further analysis. We also observe in **Fig. 9** that the impact of our new managed land mask compared to the previous one, is qualitatively similar whether it is applied to in-situ inversions or satellite inversions gridded flux fields.

Earth System
Science
Data

## 6.3 Comparsion of anthropogenic CH₄ emissions with Deng et al 2022

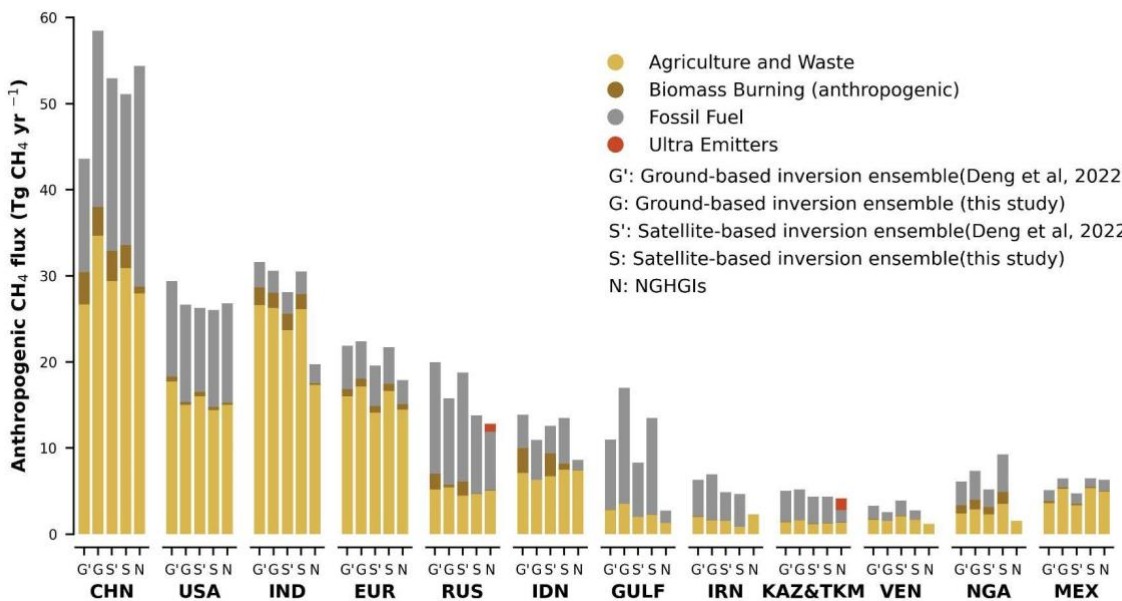

**Figure 10. Annual average of anthropogenic CH₄ emissions from in-situ (G) and satellite (S) inversions and national greenhouse gas inventories (N) during the period of 2010-2020.** G' and S' denote the anthropogenic CH₄ flux from the in-situ and satellite inversion ensembles in the previous study (Deng et al., 2022) respectively, while G and S denote the fluxes from the in-situ and satellite inversion ensembles used in this study. N denotes the estimates from NGHGIs. Grey, yellow, and brown bars represent the CH₄ fluxes from the sectors of fossil fuel combustion, agriculture and waste, and biomass burning respectively. On top of NGHGI emissions, emissions from ultra-emitters (red) are added to NGHGI estimates (diagnosed from S5P-TROPOMI measurements for the period 2019–2020; Lauvaux et al., 2022).

In our previous study, we found that satellite inversion models appear to have a better aggrement with NGHGIs than in-situ stations based inversion models, and on the other hand, that differences between inversion models and NGHGIs in large oil- and gas-producing countries suggest an underestimation of national reports, possibly due to the omission of ultra-emitting sources by NGHGIs. With the new inversion ensemble in this study, we confirm those results (**Fig 10**). In countries such as CHN, IND, and RUS, the updated inversion model set provides estimates that are closer to NGHGIs, but differences still exist, and the reasons for these differences are not the same. For example, differences in anthropogenic methane emissions in IND are mainly due to differences in agricultural and waste methane flux with the new inversion ensemble used in this study. In RUS, the updated inversion ensemble shows lower fossil fuel emissions, reducing the difference with NGHGIs for this sector, but higher agricultural and waste emissions than in Deng et al. (2022). Nevertheless, the updated fossil fuel emission flux is still higher than the NGHGIs estimate for RUS. The remaining differences may be attributed to ultra-emitting sources or underestimated emission factors for some components of the oil and gas extraction and distribution industry in RUS. Conversely, in GULF, the new inversion model ensemble consistently reflects higher fossil fuel emission fluxes than NGHGIs



like in our previous study, and expands the difference in estimates of artificial methane flux between inversion models and NGHGIs, possibly indicating more methane leakage.

## 6.4 Influence of the prior used in CH₄ inversions

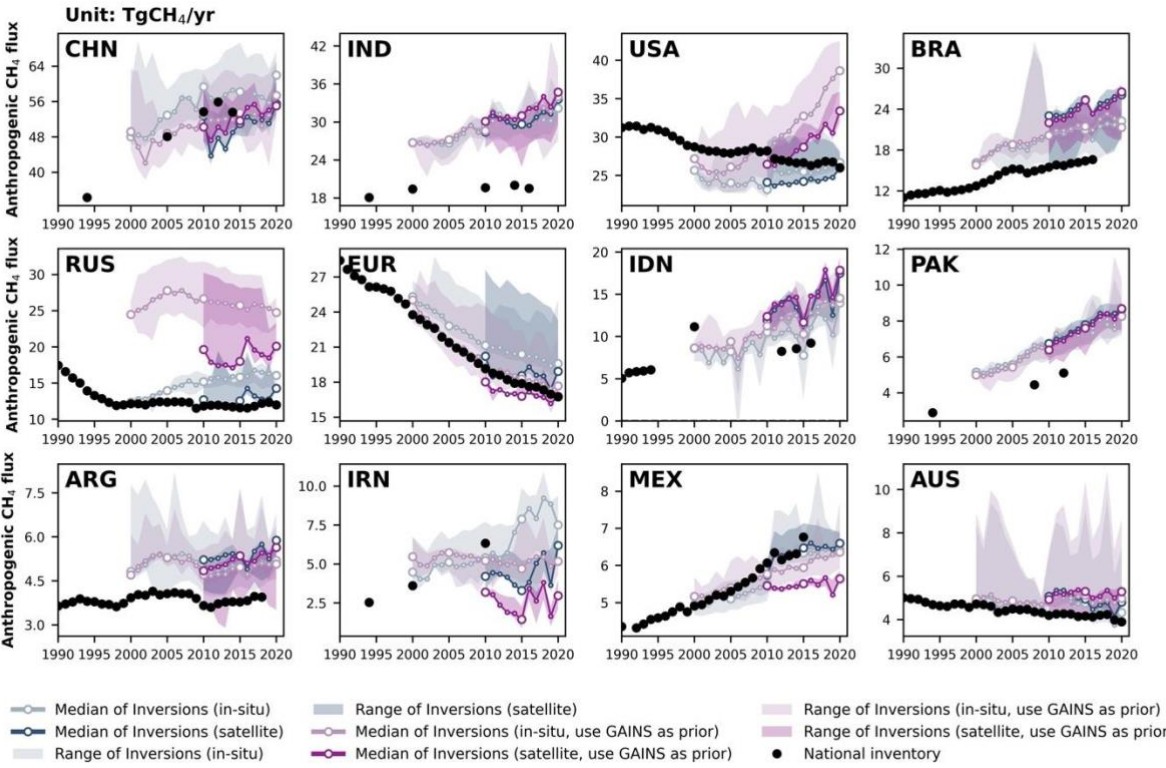

**Figure 11. Total anthropogenic CH₄ fluxes for the 12 top emitters: China (CHN), India (IND), United States (USA), Brazil (BRA), Russia (RUS), European Union (EUR), Indonesia (IDN), Pakistan (PAK), Argentina (ARG), Iran (IRN), Mexico (MEX), and Australia (AUS).** The black dots denote the reported values from NGHGIs. The light blue lines/areas denote the median and maximum-minimum ranges of in-situ CH₄ inversions based on EDGARv6.0 as the prior and the dark blue ones of satellite inversions, respectively. The light purple lines/areas denote the median and maximum-minimum ranges of in-situ CH₄ inversions based on GAINS (Höglund-Isaksson et al., 2020) as the prior and the dark purple ones of satellite inversions, respectively.

The use of different priors can also influence the inversion results of the data. **Fig 11** presents the sets of inversion results using EDGAR (blue) and GAINS (purple) as priors. In most countries, the median values of the two inversion result sets are similar. However, in countries such as RUS, USA, IRN, MEX, significant differences are observed between the two inversion result sets, which may primarily stem from the differences in the inversion results for fossil CH₄ emissions (**SI Fig 2**). In RUS and USA, the inversion results using GAINS as priors are consistently higher than those using EDGAR as priors. In RUS, the satellite inversion results using GAINS as priors are higher by 45% during 2010-2020, and the ground-based inversion results are higher by 75% during 2000-2020. In the case of the USA, the inversion results using GAINS as priors exhibit a completely





different trend compared to the ones obtained using NGHGIs and EDGAR as priors. The inversion results using GAINS as priors, both from satellite and ground-based measurements, show a rapid growth trend by increasing 24% from 2010 to 2020. In IRN and MEX, the inversion results using GAINS as priors are lower than those using EDGAR as priors. For IRN, the differences between satellite inversion results using different priors are not significant, and the trends are similar. However, the ground-based inversion results are very close between 2000-2013, but after 2013, a steep increase is observed in the ground-

based inversion results using GAINS as priors. On the other hand, in MEX, the ground-based inversion results are similar, but the satellite inversion results using GAINS as priors are relatively lower by 14% averagely.

### 6.5 Comparing anthropogenic N$_2$O flux with the previous study

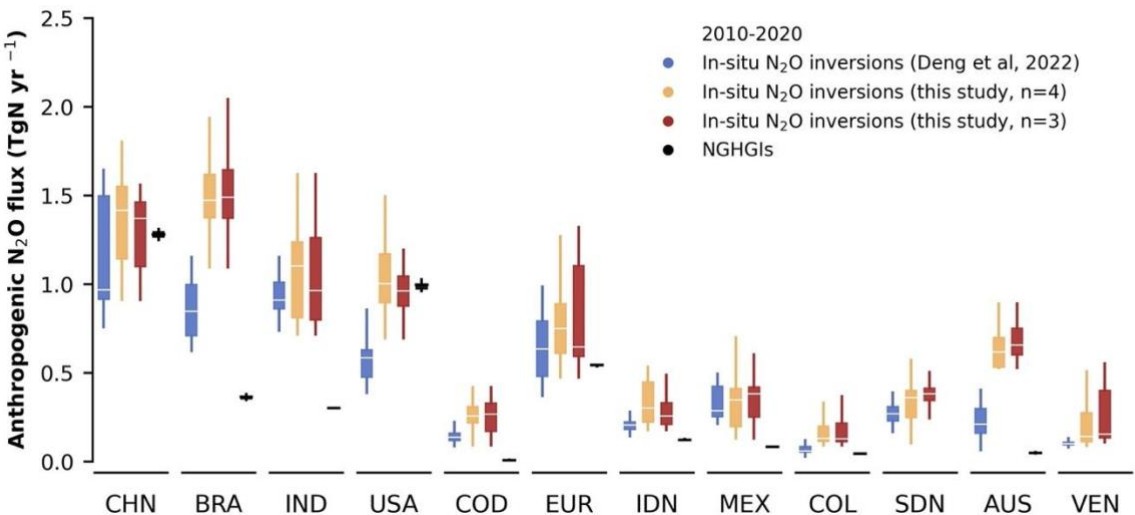

**Figure 12. Anthropogenic N$_2$O fluxes during the period of 2010-2020 in China (CHN), Brazil (BRA), India (IND), United States**
**(USA), Democratic Republic of the Congo (COD), European Union (EUR), Indonesia (IDN), Mexico (MEX), Colombia (COL), SDN (Sudan), Australia (AUS), and Venezuela (VEN).** Blue boxes denote the in-situ inversion results from Deng et al. 2022 processed from Global Carbon Budget 2020 (Friedlingstein et al., 2020). Dark yellow boxes denote the inversion results processed in this study. Black boxes denote the NGHGIs reported values.

Overall, the updated N$_2$O inversion results are systematically higher than the previous N$_2$O inversion results in most countries,
resulting in larger discrepancies between N$_2$O inversion results and NGHGIs (**Fig 12**). However, in the case of the USA, the median of our N$_2$O inversion results is very close to NGHGIs. In addition, in countries such as IND, IDN, COL, CDN, and VEN, our N$_2$O inversion results have a larger distribution compared to the previous study, indicating that the new N$_2$O inversion ensemble (n=4) has less consistency in these countries compared to the previous ensemble (n=3).





**Conclusions**

This study reconciles the gap between atmospheric inversions and UNFCCC NGHGIs for each of the three greenhouse gases, based on the post-processing framework we proposed in our previous study (Deng et al., 2022). We update inversion results and NGHGIs datasets to present the most-up-to-date discrepancies between these two estimates. For $CO_2$, we updated the inversion results up to 2021, added a new inversion ensemble including inversions based on satellite observations, and applied a new mask of national managed land based on NGHGI reports in Russia, Brazil and Canada. For $CH_4$, we compared NGHGIs

and $CH_4$ inversion results up to 2020 by splitting the anthropogenic fluxes from inversions by aggregating prior estimates from each sector or by removing fluxes of natural processes and discussed the uncertainties by using different priors in $CH_4$ inversions. For $N_2O$, we updated the inversion results up to 2019 and included the MIROC4-ACTM $N_2O$ inversion, also separated the fluxes from managed land by using the same method on $CO_2$.

In the case of $CO_2$, In this study, we updated the managed land mask for Canada, Brazil, and Russia based on maps derived

from NGHGIs and adjusted tree-cover thresholds. The analysis of different managed land mask definitions shows that the new mask, which is more consistent with the definition of managed land in the NGHGIs for these countries, improves the agreement between managed land $CO_2$ fluxes and NGHGIs in Russia and Canada. However, in Brazil, the new mask increases the gap between the estimated land carbon emissions and NGHGIs. Further analysis is needed to understand the sources and reasons for discrepancies and uncertainties between inversion estimates and NGHGIs. Thus, we still recommend that countries should

report their managed land in a spatially explicit manner to enable a better evaluation of national emission reports using inversions (and other observation-based approaches), and countries should also follow the recommendations of the IPCC 2006 Guidelines encouraging countries to use atmospheric data as an independent check on their national reports (IPCC 2006, 2019). Three additional satellite-based inversion results have been introduced for comparison with the in-situ inversion results and NGHGIs. In some countries, the satellite-based inversions demonstrate better consistency with NGHGIs compared to the in-

situ inversion models.

For $CH_4$, despite the large spread of inversions, both in-situ and GOSAT inversions show systematic differences with NGHGIs. We also found that Kazakhstan and Turkmenistan in Central Asia and the Gulf countries in the Middle East, characterized by oil- and gas-producing industries, report much less $CH_4$ emissions than atmospheric inversions estimates. While in this region, there are few ground stations, and inversions depend on their prior fluxes, the fact that GOSAT and in-situ based inversions

point to NGHGI emissions being underestimated suggests areas for future research to constrain the emissions of these countries. We recommend here to develop regional campaigns (such as those performed in Alvarez et al. (2018)), to refine emission factors, and to track regional oil, gas and coal basins emissions and ultra-emitter site-level emissions using new tools (such as moderate and high-resolution satellite imagery).

For $N_2O$, the prevalence of large tropical natural sources, being outside the responsibility of countries if they are located on

unmanaged lands, has been overlooked before. For example, nearly half of the forests in Brazil are unmanaged according to its national inventory report. We did not solve this problem, but highlighted it and proposed a new method to remove natural





emissions from inversion total emissions. As many non-Annex I countries, which will have to produce inventories for the global stocktake are tropical countries with a very active nitrogen cycle and large natural $N_2O$ emissions, a decoupling will exist between targeted emissions reductions and the observed growth rate of $N_2O$: it may hamper the eventual effectiveness of

mitigation policies, that are directly reflected in the UNFCCC NGHGIs reports, especially for this greenhouse gas. It is fair to say that the uncertainty from the spread of different inversions is large enough that inversions cannot 'falsify' N2O NGHGIs in most instances. Nevertheless, for $CH_4$ in countries around the Persian Gulf and Central Asia, and to some extent in Russia, and for $N_2O$ in tropical countries, Mexico and Australia, we found that NGHGIs emissions are significantly lower than inversions, which suggests that activity data or emission factors may need to be re-evaluated. Despite their large spread,

inversions have the advantage of providing fluxes that are consistent with the accurately observed growth rates of each greenhouse gas in the atmosphere. The uncertainty of inversions is mainly a systematic bias due to internal settings or to the choice of a transport model. It does not mean that inversions cannot be used for monitoring interannual variability and trends of fluxes, in response to mitigation efforts, since most of their bias should have a small temporal component.

The study of global inversions at the country scale rather than at the traditional subcontinent scale (e.g. the "Transcom3

regions" of Gurney et al. (2002)) obviously pushes inversions close to the limit of their domain of validity, even in the case of large countries. The densification of observation networks and systems, especially from space, increases the observational information available at all spatial scales and gradually makes it possible to study smaller countries and reduce uncertainties of inversion results. This densification must be accompanied by a corresponding increase in the horizontal resolution of inversion systems (both the transport model and the control vector to be optimized). Note that the spatial resolution of most

inverse models such as those contributing to the global carbon/methane/nitrous oxide budget is larger than 1 degree (see Table A4 in Friedlingstein et al. (2022), Table S6 in Saunois et al. (2020), and Table 1 in Tian et al. (2023)). They will likely soon have to go below one degree on a global scale to remain competitive for this type of study, despite the high computational challenge posed by the atmospheric inversion of long-lived tracers.

**Data availability**

Processed GHG ($CO_2$, $CH_4$, and $N_2O$) data from inverse models and UNFCCC NGHGIs are available at https://doi.org/10.5281/zenodo.10841716 (Deng et al., 2024).
This dataset contains 5 data files:

- The file *Inversions_CO2_v2022.csv* includes the NEE CO2 flux from managed lands for the nine CO2 inverse models. It includes 8 fields: years (from 1960 to 2021), country, value (unit: TgC/yr), sector ("land": without the

adjustment of lateral C flux; "land_cor": with later C flux adjustment), source, gas, observation ("in-situ": in-situ-based; "satellite": satellite-based), version ("CO2_ML_v2022" only).



- The file *Inversions_CH4_v2022.csv* includes CH4 flux from anthropogenic sources for the six CH4 inverse models. It includes 8 fields: years (from 2000 to 2020), country, value (unit: TgCH4/yr), sector ("agrw": agriculture and waste; "fos": fossil fuel; 'ant': anthropogenic=agrw+fos), source, gas, observation ("in-situ": in-situ-based; "satellite": satellite-based), version ("CH4_2022_V1": use EDGAR as priors; "CH4_2022_V2": use GAINS as priors).


- The file *Inversions_N2O_v2022.csv* includes the anthropogenic N2O flux from managed lands for the four N2O inverse models. It includes 8 fields: years (from 1995 to 2020), country, value (unit: TgN2O/yr), sector ("ant" only, for anthropogenic), source, gas, observation ("in-situ" only, for in-situ-based), version ("N2O_ML_v2022" only).

- The file *lateral_CO2_v2022.csv* includes the national lateral C flux from river and trade.


- The file *NGHGIs_v2022.csv* includes the national inventory data collected from UNFCCC NGHGIs (unit: Gg/yr)

**Author contribution**

PC, FC, MS, RLT, and ZD designed and coordinated the study. PC, MS, RLT, and FC designed the framework of atmosphere inversion data processing. ZD, PC, LH, MS, RLT, and FC performed the post-processing and analysis and wrote the paper. ZD, LH, and TW compiled the national greenhouse gas inventories. MS, RLT, HT, and FC gathered the global atmosphere

inversion datasets of CO2, CH4, and N2O. GG contributed the managed land mask of Brazil and Canada. FC processed the atmosphere inversion data with masks of managed lands and country boundaries. AT, SM, RJ, YN, BZ, JT, DB and AS contribute the unpublished CH4 inversion data. All authors contributed to the full text.

**Competing interests**

At least one of the (co-)authors is a member of the editorial board of Earth System Science Data.

**Acknowledgements**

The authors are very grateful to the atmosphere inversion model developers Aki Tsuruta, Arjo Segers, Bo Zheng, Chris Wilson, Christian Rödenbeck, Kelley Wells, Liesbeth Florentie, Misa Ishizawa, Naveen Chandra, Peter Bergamaschi, Prabir Patra, Shamil Maksyutov, Yi Yin, and Yosuke Niwa for the availability of their global CO$_2$, CH$_4$, and N$_2$O inversion data and acknowledge many other data providers (measurements, models, inventories, atmospheric inversions, hybrid products, etc.)

that are directly or indirectly used in this synthesis.

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
