# Peer review of "Global Greenhouse Gas Reconciliation 2022"

_Earth System Science Data, 2024_

## Referee Comment (RC1)

**Review for ESSD 2024 - 103**

**Overview:**

In this study, the authors compare an ensemble of inversion-based estimates of terrestrial greenhouse gas fluxes including $CO_2$, $CH_4$, and $N_2O$ with National Greenhouse Gas Inventory (NGHGI) estimates. The gridded inversion-based flux estimates are processed using an updated methodology so that they can be directly compared with country-(or group-of-country)-scale sector-specific flux estimates from the NGHGIs. For $CO_2$, the authors use an updated land mask to filter out inversion-based fluxes from unmanaged lands (i.e., intact forests) and also account for lateral transport of carbon via rivers, crop trade, and wood trade, allowing for direct comparison of the inversion and NGHGI non-fossil $CO_2$ fluxes. For $CH_4$, the inversion-based fluxes were partitioned into three "super-sectors" (agricultural and waste, fossil fuels, and biofuel + biomass burning excluding wildfires) for comparison with the NGHGIs. For $N_2O$, anthropogenic fluxes were obtained from the inversions by subtracting natural fluxes from lakes and rivers as well as from wildfires. For each greenhouse gas, the authors compare the ensemble of processed inversion-based flux estimates with NGHGIs in a variety of countries, while also comparing to the previous study of Deng et al (2022) to show the impact of the new datasets and updated land mask on the inversion ensembles. Notable discrepancies between the inversion ensembles and the NGHGIs are discussed for each gas. The accompanying dataset is accessible at the given identifier, is easy to use, and contains all necessary information to perform statistical tests and reproduce the key figures in the manuscript. In particular, there is nothing to indicate that the data are erroneous, and the format of the CSV files makes it very easy to perform statistical analyses on them.

I think the submitted manuscript is appropriate to support the publication of this data set. The data and methods presented are a useful and valuable update to the work presented in Deng et al (2022). The differences in methodology and data availability between this work and Deng et al (2022) are clearly explained, and their impacts on the results are illustrated throughout the manuscript by comparing the updated inversion ensembles with those of Deng et al (2022). Limitations and uncertainties associated with the methodology are also discussed in appropriate detail. The overall structure of the manuscript is sensible and easy to follow. Overall, I think this dataset and the accompanying manuscript will be useful and interesting to anyone working on global greenhouse gas emission studies in the future. I have a few general comments and suggestions for the authors to consider below, as well as a list of specific comments and technical corrections at the end of this document.

**General comments:**

The manuscript may benefit from a slightly longer discussion of the impact of the prior fluxes on the $CH_4$ inversion results in Section 6.4. The authors demonstrate that $CH_4$ emissions are sensitive to the prior in several regions in Figure 11 and SI Fig 2, and they mention this earlier in the text as well (Line 175) when they discuss the work of Tibrewal et al (2024), but I think it would helpful to expand this discussion since the impact on the results was quite pronounced in a few regions.

I also think the comparison of the $N_2O$ results between this study and Deng et al (2022) described in Section 6.5 and Figure 12 was too brief, given how much time was spent earlier in

the manuscript describing the N$_2$O methodology and results. In particular, there is a good discussion of uncertainties in anthropogenic N2O emission estimates in Section 5 where the authors explain possible reasons for the discrepancies between inversions and NGHGIs. For the sake of future users of this dataset and existing users of the Deng et al (2022) dataset, I think it would be helpful if the authors could connect some of these (or other) sources of uncertainty with the discrepancies we see in Figure 12.

The accompanying dataset may benefit from some additional metadata, specifically for flux units and full "super-sector" names. This information is all available in the manuscript, but it would be more convenient for future users of these files if all the necessary information was self-contained in the data files. If this is not practical, perhaps include a README file with additional metadata on the Zenodo archive.

The presentation quality of the manuscript and dataset is generally good. The methods section in particular was very thorough. However, some of the descriptions of results in Sections 3, 4.1, and 4.2 are a bit too long in my opinion. Figures 3, 4, 5, and 6 are well-made and easy to understand, so some of the descriptions of these results could probably be made a bit more concise. Shortening these descriptions would also free up some space to expand on the discussion in Sections 6.4 and 6.5, as mentioned above. Other than a few typos or small grammatical issues, the language was consistent and precise. Tables were informative and easy to read, and Figures were well-made and clear (except colour issue in Figure 3, see "Minor comments").

I found it difficult to keep track of all the abbreviations used throughout the paper. There are dozens of abbreviations in the text (including subscripts for flux terms, countries or groupings of countries, models used in different inversions, and different kinds of reports for the NGHGIs), and I often found myself flipping back to earlier pages to remind myself what some of them meant. I appreciated that country abbreviations were always re-stated in the figure captions; perhaps periodically re-stating some of the other definitions would be useful as well.

**Specific comments:**
*Lines 57 – 58:*
This sentence is unclear to me. The equivalent sentence on Zenodo makes more sense: "Much denser sampling of atmospheric CO2 and CH4 concentrations by different satellites, coordinated into a global constellation, is expected in the coming years."

*Lines 107 – 113:*
The formatting here is a bit awkward to read, specifically because of the additional semicolons in points 2 and 3. Maybe list these as 6 questions instead of 4, or find a way to rephrase questions 2 and 3.

*Table 1:*
Could you add a column to this table specifying which transport model is used by each CO2 inversion system? The format of Table 2 and Table 3 is a bit different and the inversion names

seem to include more information about the transport models used (e.g., TM5-CAMS, GEOS-Chem, etc.).

*Lines 175 – 176:*
Missing a reference for GAINS, and this sentence is incomplete.

*Lines 207 – 209:*
Is it reasonable to consider everything except intact forests as "managed land"? What about other natural landcover types such as arctic tundra or grasslands (outside of Russia, where you assumed they were all managed)? Would excluding these other types of landcover from the Net Ecosystem Exchange CO2 Flux make a significant difference in your comparison between the inversions and NGHGI estimates?

*Line 306:*
The definition for $E_{nat}^{aq}$ in Equation 4 is not given until Line 321. Can you move the explanation closer to Equation 4 so that the reader does not have to skip ahead to see what the terms mean?

*Lines 352 – 357:*
This explanation of how you compared the Net Land CO2 Flux is clear and reasonable, but it would be helpful if you could mention this earlier. In Equation 1 and Line 231 you explain that you are comparing the adjusted inversion NEE flux with the anthropogenic NGHGI fluxes, but it's a bit ambiguous because the adjusted inversion NEE flux does not include fossil CO2 whereas the anthropogenic NGHGI flux does. So it would be helpful to emphasize earlier in the text that the comparison in Equation 1 will actually be done for only the non-fossil component of the NGHGI fluxes.

*Figure 3:*
It is very hard to see the difference between the light green shading (range of in-situ inversions) and dark green shading (range of satellite inversions). Can you adjust the colour scheme? It was easier to see the different shades in the CH4 and N2O figures so I don't think you need to change those ones.

*Lines 410 – 420 (Figure 3 caption):*
Can you briefly mention the issue with switching from GOSAT to OCO-2 (described in detail on Lines 464 – 470) in this Figure caption? This is crucial for interpreting the time series plots, so it would be helpful to quickly mention that the satellite inversions use only OCO-2 after 2015 in the caption.

*Line 565:*
The acronym "IAV" is not defined in the text; is this interannual variability?

*Line 641:*
Can you give your definition of "ultra-emitter" and explain why they are omitted from NGHGIs?

*Lines 820 - 841:*

Consider moving SI Fig 2 to the main text to help illustrate impact of prior emissions (see also my general comment about expanding section 6.4).

**Technical corrections:**

*Line 90:*
"our framework to process inversion" --> "our framework to process the inversions"

*Line 92:*
"Atmosphericinversions" --> "Atmospheric inversions"

*Line 385:*
"Indonesia (IND)" --> "Indonesia (IDN)"

*Line 391:*
"is a large emitter of oil and gas" --> "is a large producer of oil and gas"

*Line 441:*
"that the NGHGIs reports" --> "than the NGHGIs reports"

*Lines 443 – 445:*
The font here is different from the rest of the text.

*Line 446:*
Replace semicolon with period.

*Line 718:*
"which may underestimate emissions n when soil" --> ?? I am not sure if there is a missing word here, or if you should just delete the "n"

*Line 768:*
"difference" --> "differences"

*Line 864:*
Delete "In this study,"

---

## Author Comment (AC1)

Reply to RC1: 'Comment on essd-2024-103', Christian DiMaria, 29 Jul 2024

Review for ESSD 2024 - 103
Overview:
In this study, the authors compare an ensemble of inversion-based estimates of terrestrial greenhouse gas fluxes including CO2, CH4, and N2O with National Greenhouse Gas Inventory (NGHGI) estimates. The gridded inversion-based flux estimates are processed using an updated methodology so that they can be directly compared with country-(or group-of-country)-scale sector-specific flux estimates from the NGHGIs. For CO2, the authors use an updated land mask to filter out inversion-based fluxes from unmanaged lands (i.e., intact forests) and also account for lateral transport of carbon via rivers, crop trade, and wood trade, allowing for direct comparison of the inversion and NGHGI non-fossil CO2 fluxes. For CH4, the inversion-based fluxes were partitioned into three "super-sectors" (agricultural and waste, fossil fuels, and biofuel + biomass burning excluding wildfires) for comparison with the NGHGIs. For N2O, anthropogenic fluxes were obtained from the inversions by subtracting natural fluxes from lakes and rivers as well as from wildfires. For each greenhouse gas, the authors compare the ensemble of processed inversion-based flux estimates with NGHGIs in a variety of countries, while also comparing to the previous study of Deng et al (2022) to show the impact of the new datasets and updated land mask on the inversion ensembles. Notable discrepancies between the inversion ensembles and the NGHGIs are discussed for each gas. The accompanying dataset is accessible at the given identifier, is easy to use, and contains all necessary information to perform statistical tests and reproduce the key figures in the manuscript. In particular, there is nothing to indicate that the data are erroneous, and the format of the CSV files makes it very easy to perform statistical analyses on them.
I think the submitted manuscript is appropriate to support the publication of this data set. The data and methods presented are a useful and valuable update to the work presented in Deng et al (2022). The differences in methodology and data availability between this work and Deng et al (2022) are clearly explained, and their impacts on the results are illustrated throughout the manuscript by comparing the updated inversion ensembles with those of Deng et al (2022). Limitations and uncertainties associated with the methodology are also discussed in appropriate detail. The overall structure of the manuscript is sensible and easy to follow. Overall, I think this dataset and the accompanying manuscript will be useful and interesting to anyone working on global greenhouse gas emission studies in the future. I have a few general comments and suggestions for the authors to consider below, as well as a list of specific comments and technical corrections at the end of this document.

Thank you for the positive and very constructive comments. Please find below our replies.

General comments:
The manuscript may benefit from a slightly longer discussion of the impact of the prior fluxes on the CH4 inversion results in Section 6.4. The authors demonstrate that CH4 emissions are sensitive to the prior in several regions in Figure 11 and SI Fig 2, and they mention this earlier in the text as well (Line 175) when they discuss the work of Tibrewal et al (2024), but I think it

would helpful to expand this discussion since the impact on the results was quite pronounced in a few regions.

We appreciate the reviewer's comment. We have emphasized that the differences in prior fluxes introduce additional uncertainties into the inversion results. These discrepancies arise from the varying methodologies and data sources used in the inventories used as priors. Specifically, the two priors included in this study, EDGAR and GAINS, have distinct foundations; EDGAR is based on IPCC emission factors and IEA energy statistics, while GAINS employs an independent methodology. As a result, the estimates derived from these two inventories inherently differ.

In response to your suggestion, we have expanded the discussion in Section 6.4 to better articulate the impact of these prior fluxes on the CH4 inversion results. Additionally, we have moved Figure S2 to the main text, now labeled as Figure 12, to provide a clearer visual representation of the sensitivity of CH4 emissions to these prior fluxes in various regions.

**Line 881-888**:

*Such discrepancies may arise from differences in inventory methodologies and the resulting estimations. As shown in Supplementary Figure S1 in Tibrewal et al. (2024), similar discrepancies were found between the two inventories in these countries, which reports a higher estimation from GAINS in RUS and USA compared to EDGAR during 2011-2020, and a lower estimation in IRN. As noted in Tibrewal et al. (2024), EDGAR is based on various versions of National Inventory Reports (NIR) that utilize different combinations of emission factors from the IPCC, while GAINS employs an independent estimation approach. This highlights the critical role of prior data selection in determining the accuracy of $CH_4$ emission estimates.*

[Figure]

*Fig 12. Annual average of anthropogenic $CH_4$ emissions from in-situ and satellite inversions based on two different priors during the period of 2010-2020. GE and SE denote the anthropogenic $CH_4$ flux from the in-situ and satellite inversion ensembles based on*

*EDGARv6.0 as the prior, while GG and SG represent the in-situ and satellite CH$_4$ inversions based on GAINS as the prior.*

I also think the comparison of the N2O results between this study and Deng et al (2022) described in Section 6.5 and Figure 12 was too brief, given how much time was spent earlier in the manuscript describing the N2O methodology and results. In particular, there is a good discussion of uncertainties in anthropogenic N2O emission estimates in Section 5 where the authors explain possible reasons for the discrepancies between inversions and NGHGIs. For the sake of future users of this dataset and existing users of the Deng et al (2022) dataset, I think it would be helpful if the authors could connect some of these (or other) sources of uncertainty with the discrepancies we see in Figure 12.

Thanks for your comment. We extent the discussion in Section 6,5:

**Line 910-923***: The updated N2O inversion results show systematically higher anthropogenic emissions than the previous N2O inversion results (Deng et al., 2022), resulting in larger discrepancies between N2O inversion results and NGHGIs in most countries, as illustrated in Figure 13. Countries such as Brazil (BRA), Democratic Republic of the Congo (COD), Indonesia (IDN), Colombia (COL), Sudan (SDN), Australia (AUS), and Venezuela (VEN) exhibit significant differences. These discrepancies may be attributed to the use of lower IPCC default emission factors in the national inventories of these tropical countries, leading to lower NGHGI results. The IPCC default emission factors are derived from measurements primarily conducted in temperate regions of the Northern Hemisphere (e.g., Europe and the USA), which explains the better alignment of inversion results with inventories in those regions. Notably, in the case of the USA, the median of the updated N2O inversion results is very close to the NGHGIs. The median of the N2O inversion results from Deng et al. (2022) was 42% lower than the NGHGIs between 2005 and 2015, whereas the median of the updated inversion models is only 4% lower. This demonstrates improved consistency in the updated inversion system results for the USA. Additionally, in countries such as India (IND), Indonesia (IDN), Colombia (COL), Canada (CAN), and Venezuela (VEN), our N2O inversion results show a larger distribution compared to the previous study. This indicates that the new N2O inversion ensemble (n=4) has less consistency in these countries compared to the previous ensemble (n=3).*

The accompanying dataset may benefit from some additional metadata, specifically for flux units and full "super-sector" names. This information is all available in the manuscript, but it would be more convenient for future users of these files if all the necessary information was self-contained in the data files. If this is not practical, perhaps include a README file with additional metadata on the Zenodo archive.

Thank you for your insightful suggestion. We agree that including additional metadata, such as flux units and full "super-sector" names, would enhance the usability of the dataset. We've added a README file with the necessary metadata and will make it available in the Zenodo archive (https://doi.org/10.5281/zenodo.13887128).

The presentation quality of the manuscript and dataset is generally good. The methods section in particular was very thorough. However, some of the descriptions of results in Sections 3, 4.1, and 4.2 are a bit too long in my opinion. Figures 3, 4, 5, and 6 are well-made and easy to understand, so some of the descriptions of these results could probably be made a bit more concise. Shortening these descriptions would also free up some space to expand on the discussion in Sections 6.4 and 6.5, as mentioned above. Other than a few typos or small grammatical issues, the language was consistent and precise. Tables were informative and easy to read, and Figures were well-made and clear (except colour issue in Figure 3, see "Minor comments").

We appreciate your feedback regarding the length of the descriptions for each selected country. Our intention was to fully showcase and discuss the specific situations and differences among these countries. In response to your comments, we have balanced the manuscript by condensing the sections as suggested while also expanding Sections 6.4 and 6.5 for a more comprehensive discussion. Additionally, we have corrected any spelling errors, grammatical issues, and the color problem in Figure 3.

I found it difficult to keep track of all the abbreviations used throughout the paper. There re dozens of abbreviations in the text (including subscripts for flux terms, countries or groupings of countries, models used in different inversions, and different kinds of reports for the NGHGIs), and I often found myself flipping back to earlier pages to remind myself what some of them meant. I appreciated that country abbreviations were always re-stated in the figure captions; perhaps periodically re-stating some of the other definitions would be useful as well.

Thanks for the suggestion. We re-state the country abbreviations when they are first mentioned in every Section. And we also add a glossary of country abbreviations in **Table S1**:

*Table S1. Glossary of country abbreviations.*

| Gas | Super Sector | Country List |
|---|---|---|
| $CO_2$ | *Net Land Flux* | **AUS**: Australia,
**BRA**: Brazil,
**CAN**: Canada,
**CHN**: China,
**COD**: Democratic Republic of the Congo,
**EUR**: European Union,
**IND**: India,
**KAZ**: Kazakhstan,
**MNG**: Mongolia,
**RUS**: Russia,
**USA**: United States,
**ZAF**: South Africa |
| $CH_4$ | *Anthropogenic* | **ARG**: Argentina,
**AUS**: Australia,
**BRA**: Brazil, |

|  |  | **CHN**: China,
**EUR**: European Union,
**IDN**: Indonesia,
**IND**: India,
**IRN**: Iran,
**MEX**: Mexico,
**PAK**: Pakistan,
**RUS**: Russia,
**USA**: United States |
|  | *Fossil* | **CHN**: China,
**EUR**: European Union,
**GULF**: Gulf countries (Saudi Arabia + Oman + United Arab Emirates + Kuwait + Bahrain + Iraq + Qatar),
**IDN**: Indonesia,
**IND**: India,
**IRN**: Iran,
**KAZ&TKM**: Kazakhstan + Turkmenistan,
**MEX**: Mexico,
**NGA**: Nigeria,
**RUS**: Russia,
**USA**: United States,
**VEN**: Venezuela |
|  | *Agriculture and Waste* | **ARG**: Argentina,
**BGD**: Bangladesh,
**BRA**: Brazil,
**CHN**: China,
**EUR**: European Union,
**IDN**: Indonesia,
**IND**: India,
**MEX**: Mexico,
**PAK**: Pakistan,
**RUS**: Russia,
**THA**: Thailand,
**USA**: United States |
| $N_2O$ | *Anthropogenic* | **AUS**: Australia,
**BRA**: Brazil,
**CHN**: China,
**COD**: Democratic Republic of the Congo,
**COL**: Columbia,
**EUR**: European Union,
**IDN**: Indonesia,
**IND**: India,
**MEX**: Mexico,
**SDN**: Sudan,
**USA**: United States,
**VEN**: Venezuela |

Specific comments:
Lines 57 – 58:
This sentence is unclear to me. The equivalent sentence on Zenodo makes more sense: "Much denser sampling of atmospheric CO2 and CH4 concentrations by different satellites, coordinated into a global constellation, is expected in the coming years."

Thank you for the comment. We revised the sentence:

**Line 57-58**: *Much denser sampling of atmospheric CO2 and CH4 concentrations by different satellites, coordinated into a global constellation, is expected in the coming years.*

Lines 107 – 113:
The formatting here is a bit awkward to read, specifically because of the additional semicolons in points 2 and 3. Maybe list these as 6 questions instead of 4, or find a way to rephrase questions 2 and 3.

Thank you for your suggestion. We have revised the four questions into six for improved clarification:

**Line 110-117**: *Based on the newly updated inversion results and inventory, and an improvement in the methodology framework proposed in the previous study (Deng et al., 2022), we specifically address the following questions: 1) how do inversion models compare with NGHGIs for the three gases?; 2) what are the plausible reasons for mismatches between inversions and NGHGIs?; 3) did the new maps of managed land masks in this study reduce the mismatch between the inversions and NGHGIs for CO2 and N2O?; 4) what independent information can be extracted from inversions to evaluate the mean values or the trends of greenhouse gas emissions and removals?; 5) does this information exhibit a good agreement with NGHGIs?; and 6) how do satellite-retrieval driven inversion models differ from the surface in-situ and flask sampling driven inversion model results?*

Table 1:
Could you add a column to this table specifying which transport model is used by each CO2 inversion system? The format of Table 2 and Table 3 is a bit different and the inversion names seem to include more information about the transport models used (e.g., TM5-CAMS, GEOS-Chem, etc.).

Thank you for your suggestion. We have added a column in Table 1 to specify the transport model used by each CO2 inversion system.

**Table 1**:

| Inversion System | Version | Period | Observation | Transport Model |
|---|---|---|---|---|
| CarbonTracker Europe (CTE): CTE2022_SiB4 (van der Laan-Luijkx et al., 2017) | v2022 | 2001-2021 | Ground-based Obspack GLOBALVIEW plus v7.0 and NRT_v7.2 | TM5 |
| Jena Carboscope sEXTocNEET (Rödenbeck et al., 2003) | v2022 | 1960-2021 | | TM3 |
| Copernicus Atmosphere Monitoring Service (CAMS) (Chevallier et al., 2005) | v21r1 | 1979-2021 | | LMDZ v6 |
| The University of Edinburgh (UoE) (Feng et al., 2016) | v6.1b | 2001-2021 | | GEOS-CHEM |
| the NICAM-based Inverse Simulation for Monitoring CO2 (NISMON-CO2) (Niwa et al., 2022) | v2022.1 | 1990-2021 | | NICAN-TM |
| CMS-Flux (Liu et al., 2021), | v2022 | 2010-2021 | Ground-based & ACOS-GOSAT v9; OCO-2 v10 scaled to WMO2019 | GEOS-CHEM |
| CAMS-Satellite (Chevallier et al., 2005) | FT21r2 | 2010-2021 | bias-corrected ACOS GOSAT v9 over land until August 2014 + bias-corrected ACO S OCO-2 v10 over land, both rescaled to WMO2019 | LMDZ v6 |
| THU (Kong et al., 2022) | v2022 | 2015-2021 | OCO-2 v10r data scaled to WMO2019 | GEOS-CHEM |
| GONGGA (Jin et al., 2023) | v2022 | 2015-2021 | OCO-2 v10r data scaled to WMO2019 | GEOS-CHEM |

Lines 175 – 176:
Missing a reference for GAINS, and this sentence is incomplete.

We've added the reference to GAINS and completed the sentence:

**Line 181-183**: *During the production of the inversion simulations, GAINS inventory (Höglund-Isaksson, 2020) was proposed to use another prior for fossil fuel sources instead of using EDGAR v6.*

Lines 207 – 209:
Is it reasonable to consider everything except intact forests as "managed land"? What about other natural landcover types such as arctic tundra or grasslands (outside of Russia, where you assumed they were all managed)? Would excluding these other types of landcover from the Net Ecosystem Exchange CO2 Flux make a significant difference in your comparison between the inversions and NGHGI estimates?

Thanks for the comment. The NGHGIs encompass all managed land types, including forest land, cropland, grassland, wetlands, settlements, and other areas and transitions of land use between them. While using intact forests as a mask for unmanaged land does overlook other natural land cover types, such as arctic tundra and grasslands, it is important to note, as Grassi et al. (2021) indicate, that the majority of CO2 fluxes in NGHGIs are attributed to forests land (Paragraph 3 in Section '*Country data submitted to UNFCCC*'): "In principle, LULUCF includes all land uses (forest land, cropland, grassland, wetlands, settlements and other land). In practice, although almost all Annex I countries report all land uses, many non-Annex I countries report only on forest land and deforestation (that is, forest converted into other land uses). When splitting the net LULUCF flux into the three categories 'forest', 'deforestation' and 'other' (which includes peat decomposition in Indonesia), forest and deforestation represent more than 95% of the sum of absolute fluxes from the three categories in both Annex I and non-Annex I countries (Supplementary Section 2). This does not mean that other fluxes are unimportant—for example, emissions from agricultural organic soils or removals from grasslands may be relevant in some countries—but simply that at the global level their net sum is close to zero in NGHGIs. In Fig. 1b, 'emissions' include fluxes from deforestation and, for Indonesia, peat fires; 'removals' include fluxes from forest land and from 'other'."

We have also acknowledged the limitations of using intact forests as a proxy for unmanaged land in our manuscript. Specifically, we state in **Line 225-228**: "*This approach assumes that non-intact forest areas can serve as a reasonably good proxy for managed forests reported in the NGHGIs (Grassi et al., 2021, 2023). It is important to note that this approach is somewhat arbitrary, as highlighted in previous studies (Ogle et al., 2018; Chevallier, 2021; Grassi et al., 2021). However, in the absence of a machine-readable definition of managed plots in many NGHGIs, there is currently no better alternative available.*"

Line 306:
The definition for ___ in Equation 4 is not given until Line 321. Can you move the explanation closer to Equation 4 so that the reader does not have to skip ahead to see what the terms mean?

Thanks, we've added a explanation of $E_{nat}^{aq}$ and $E_{ant}^{ni}$ below Equation 4.

**Line 322-323**:

$$E_{ant}^{inv} = E_{ML}^{inv} - E_{nat}^{aq} - E_{wildfires}^{GFED} \Leftrightarrow E_{ant}^{ni} \tag{4}$$

Here, the natural N$_2$O sources include natural emissions from freshwater systems ($E_{nat}^{aq}$) and natural emissions from wildfires ($E_{ant}^{ni}$).

This explanation of how you compared the Net Land CO2 Flux is clear and reasonable, but it would be helpful if you could mention this earlier. In Equation 1 and Line 231 you explain that you are comparing the adjusted inversion NEE flux with the anthropogenic NGHGI fluxes, but it's a bit ambiguous because the adjusted inversion NEE flux does not include fossil CO2 whereas the anthropogenic NGHGI flux does. So it would be helpful to emphasize earlier in the text that the comparison in Equation 1 will actually be done for only the non-fossil component of the NGHGI fluxes.

Thank you for the suggestion. We revised the corresponding sentences to clarify the comparison is actually done for only the non-fossil component of CO$_2$ flux.

**Line 230-240**: *In addition to the extraction of fossil CO$_2$ flux and managed land CO$_2$ flux, there are CO$_2$ fluxes that are part of $F_{ML}^{inv\,NEE}$ but are not counted by NGHGIs. These fluxes are induced by (i) soils to rivers to oceans carbon export ($F_{ML}^{rivers}$) which has an anthropogenic and a natural component (Regnier et al., 2013), and (ii) net anthropogenic export of crop and wood products across each country's boundary ($F_{ant}^{crop\,trade}$ and $F_{ant}^{wood\,trade}$). The magnitudes of these CO$_2$ fluxes are different between countries, and values from the selected countries are presented in **SI Fig 1**. We assume that NGHGIs include CO$_2$ losses from fire (wildfire and prescribed fire) and other disturbances (wind, pests) and from domestic harvesting, as recommended by the IPCC reporting guidelines (IPCC, 2006, 2019) (although some countries, such as Canada and Australia exclude some emissions from these disturbances, and the subsequent removals from the same areas (Grassi et al., 2023)). The adjusted inversion NEE that can be compared with inventories, $F_{adj}^{inv\,NEE}$, is given by:*

$$F_{adj}^{inv\,NEE} = F_{ML}^{inv\,NEE} - F_{ML}^{rivers} - F_{ant}^{crop\,trade} - F_{ant}^{wood\,trade} \quad \Leftrightarrow F_{ant-nf}^{ni},$$
$$(1)$$

*where the sign $\Leftrightarrow$ means 'compared with', $F_{ant-nf}^{ni}$ is the non-fossil part of the anthropogenic CO$_2$ flux from NGHGIs, $F_{tot}^{rivers}$ is the sum of the natural and anthropogenic CO$_2$ flux on land from CO$_2$ fixation by plants that is leached as carbon via soils and channeled to inland waters to be exported to the ocean or to another country.*

It is very hard to see the difference between the light green shading (range of in-situ inversions) and dark green shading (range of satellite inversions). Can you adjust the colour scheme? It was easier to see the different shades in the CH4 and N2O figures so I don't think you need to change those ones.

We've adjusted the color scheme in Fig 3.
**Fig 3**:

[Figure]

Lines 410 – 420 (Figure 3 caption):
Can you briefly mention the issue with switching from GOSAT to OCO-2 (described in detail on Lines 464 – 470) in this Figure caption? This is crucial for interpreting the time series plots, so it would be helpful to quickly mention that the satellite inversions use only OCO-2 after 2015 in the caption.

Thank you for your suggestion. Based on your input, we have added the following sentence to the **Figure 3 caption**: *Otherwise, when the inversion sets have incomplete inverse models within the time interval (in-situ: n<5; satellite: n<4), their median values are represented as dashed green lines. Besides, before 2015, only GOSAT was available for the 2 of 4 satellite-based inversions, until September 2014 when the OCO-2 record started. The shading area denotes the min-max range of inversions. The red dashed lines denote the median of inversions presented by the previous study (Deng et al., 2022).*

Line 565:
The acronym "IAV" is not defined in the text; is this interannual variability? that the NGHGIs reports

Thank you for your suggestion. In the revised manuscript, we have replaced "IAV" with "interannual variability".

Line 641:
Can you give your definition of "ultra-emitter" and explain why they are omitted from NGHGIs?

Thanks for the comment. According to Lauvaux et al. (2022), "ultra-emitters" are defined as short-duration leaks from oil and gas facilities (e.g., wells, compressors) with individual emissions exceeding 20 t CH4 per hour, typically lasting less than one day. Such leaks are often random occurrences and difficult to quantify, which is why most countries do not account for these significant and episodic events in their national inventories. Consequently, they are not reported in NGHGIs.

**Line 672-674**: *The ultra-emitters defined by Lauvaux et al. (2022) are namely all short-duration leaks from oil and gas facilities (e.g., wells, compressors) with an individual emission >20 t CH4/h, each event lasting generally less than one day. Such leaks are often random occurrences and difficult to quantify, which is why most countries do not account for these significant and episodic events in the national inventories.*

Lines 820 - 841:
Consider moving SI Fig 2 to the main text to help illustrate impact of prior emissions (see also my general comment about expanding section 6.4).

Thank you for your suggestion. We have converted SI Fig 2 into Figure 12 and moved it to Section 6.4. We have also expanded the illustration in Section 6.4.

Technical corrections:
Line 90:
"our framework to process inversion" --> "our framework to process the inversions"

Line 92:
"Atmosphericinversions" --> "Atmospheric inversions"

Line 385:
"Indonesia (IND)" --> "Indonesia (IDN)"

Line 391:
"is a large emitter of oil and gas" --> "is a large producer of oil and gas"

Line 441:
"that the NGHGIs reports" --> "than the NGHGIs reports"

Lines 443 – 445:

The font here is different from the rest of the text.

Line 446:
Replace semicolon with period.

Line 718:
"which may underestimate emissions n when soil" --> ?? I am not sure if there is a missing word here, or if you should just delete the "n"

Line 768:
"difference" --> "differences"

Line 864:
Delete "In this study,"

Thank you for the comments. We've corrected the technical issues accordingly.

---

## Author Comment (AC2)

Reply to RC2: 'Comment on essd-2024-103', Anonymous Referee #2, 06 Sep 2024

*Review of Global "Greenhouse Gas Reconciliation 2022"*

Summary- In this manuscript, authors present an updated dataset based on Deng et al (2022) which presents a dataset that can be used to compare GHG emissions from national inventories to those based on model ensembles. Specifically, their method (amongst other things) uses inversions of modelled estimates from ensembles to reconcile the top-down approach taken by models with the bottom-up inventories. This paper is well written and clearly an important contribution to the literature. I recommend publication after minor revisions. Also, I apologize to the authors and the editors for my delayed review (I was on leave and could not get to this).

Thank you for the positive and very constructive comments. Please find below our replies.

Comments-

1. National LUC CO2 emissions/uptake- From the manuscript, I gathered that the authors have used the LUC emissions/uptake from the Global Carbon Project (specifically atmospheric inversions of the LUC emissions/uptake data). However, starting recently, the GCP has been updated to provide national inventories of LUC emissions and uptake (See the paper from Gasser et al which introduced this-https://bg.copernicus.org/articles/17/4075/2020/). This nationalized data is only available from 3 models. But the reason its important to discuss this is because the Gasser et al. work was done specifically to compare LUC emissions data to national inventories. Can the authors compare the national CO2 data from inversions to the nationalized data? By the way the data itself is available here- https://www.icos-cp.eu/science-and-impact/global-carbon-budget/2022 (See the third spreadsheet). I believe this will add more robustness to the validation.

    Thank you for your detailed feedback. In our study, we primarily compared the top-down atmospheric inversion models with bottom-up national inventory data to explore the methodological framework recommended by the IPCC and the associated uncertainties in the context of global accounting. Regarding the differences between global bookkeeping models and national inventories, previous studies by Grassi et al. (2018, 2021, 2023) have addressed these comparisons and analyzed the sources of discrepancies ( mainly definitions ). Specifically, aside from inconsistencies in the definitions of managed forest CO2 fluxes, the indirect effects of anthropogenic environmental changes (e.g., the CO2 fertilization effect on vegetation growth and carbon sinks due to increased atmospheric CO2) are treated as non-anthropogenic factors in bookkeeping models, while most national inventories categorize them as anthropogenic. The adjusted atmospheric inversion results in our study are closer in

definition to national inventories since the indirect anthropogenic influences have not been separated out.

While we consider that a comparison with bookkeeping models extends beyond the scope of our discussion for the reasons mentioned above, we appreciate the reviewer's suggestion. We have added a paragraph in the main text for further discussion and included a figure in the supplementary materials (building on Fig 8 to include bookkeeping model results) to briefly describe the differences among the three approaches.

**Line 766-783**:

*Figure 8. Net CO2 land fluxes during the period of a) 2011-2015; and b) 2016-2020 in China (CHN), United States (USA), European Union (EUR), Russia (RUS), Canada (CAN), Kazakhstan (KAZ), Mongolia (MNG), India (IND), Brazil (BRA), Democratic Republic of the Congo (COD), South Africa (ZAF), and Australia (AUS). Blue boxes denote the in-situ inversion results from Deng et al. (2022) processed from Global Carbon Budget 2020 (Friedlingstein et al., 2020). Light green boxes denote the in-situ inversion results processed in this study, while dark green boxes denote the satellite inversion results. Black boxes denote the NGHGIs reported values. The white lines in the boxes denote the medians of the land CO2 fluxes. Note that the inversion results here have been adjusted by the lateral flux before the comparison. Additionally, we extend the comparison with national land use change emissions from global bookkeeping models in Fig S4. Further differences between bookkeeping models limited to land use change fluxes and national inventories which additionally account for sinks in managed forests as explained in Grassi et al.*

**Fig S2** in supplementary materials:

[Figure]

*Fig S4. Net CO2 land fluxes during the period of a) 2011-2015; and b) 2016-2020 in China (CHN), United States (USA), European Union (EUR), Russia (RUS), Canada (CAN), Kazakhstan (KAZ), Mongolia (MNG), India (IND), Brazil (BRA), Democratic Republic of the Congo (COD), South Africa (ZAF), and Australia (AUS). The red boxes denote the national land use change emissions from three global bookkeeping models (i.e., BLUE, H&N2017 and OSCAR) from (Friedlingstein et al., 2023) Other legends are same as Fig 8.*

**References**:

Grassi, Giacomo, et al. "Reconciling global-model estimates and country reporting of anthropogenic forest CO2 sinks." *Nature Climate Change* 8.10 (2018): 914-920.

Grassi, Giacomo, et al. "Critical adjustment of land mitigation pathways for assessing countries' climate progress." *Nature Climate Change* 11.5 (2021): 425-434.

Grassi, Giacomo, et al. "Harmonising the land-use flux estimates of global models and national inventories for 2000–2020." *Earth System Science Data* 15.3 (2023): 1093-1114.

2. CH4 inversions- For the CH4 inversions, the authors suggest that some inversions optimize within sectors while others provide total gridded emissions. When total gridded emissions are available, prior fluxes are used to allocate emissions to sectors. Could you elaborate which inversions were differentiated by sectors and which were not? Also does assigning sectoral information based on priors involve any uncertainty? Perhaps

this was discussed in the previous paper already and this paper just needs to mention that. But regardless, a discussion of this point would be helpful.

Thanks for the comment. In this study, we utilized seven inversion systems, of which two (MIROC4-ACTM and TM5-CAMS) only provide total emissions. Assigning sectoral information based on prior fluxes does indeed introduce uncertainty, which we emphasize in Section 2.4 of our paper (Line 289-292). We have modified the text accordingly to clarify which inversion systems do not differentiate by sector and to highlight the potential uncertainties associated with using prior information for sectoral allocation:

**Line 183-188**: *Some inversions optimize emissions in groups of sectors, and others only provide total gridded emissions (MIROC4-ACTM and TM5-CAMS, detailed table can be found in Table S10 in Saunois et al, 2024). For the latter, we computed the emission from each sector within each pixel based on the proportion of the prior fluxes. Such processing can lead to significant uncertainties if not all sources increase or change at the same rate in a given region/pixel.*

3. Wood fuel burning vs fire – I appreciate the discussion by the authors when it comes to discussing the limitations when separating out the emissions from fire vs those from regular wood fuel burning. However, there has been some work recently to separate out wood fuel burning emissions which are non-renewable. Specifically, this paper- https://essd.copernicus.org/articles/15/2179/2023/essd-15-2179-2023.html. Can the authors compare their wood fuel emissions to the emissions as shown in the article here? Once again, this would make the results more robust more than anything else.

Thank you for the suggestion. We compared wood fuel emissions from Flammi et al. (2023) with our adjusted methane inversion results for biomass and biofuel burning (see the figure below), and found that the former generally underestimates the latter. Flammi et al. used wood fuel consumption data from the UNSD Energy Statistics database to provide bottom-up estimates of CO2, CH4, and N2O emissions for 125 countries. Their calculations focus on the non-renewable biomass fraction in cooking, which is country-specific, with an average of 42% across 91 countries.

In our study, as we stated in Line 307-314, the biomass and biofuel burning (BB) inversion category includes methane emissions from wildfires in forests, savannahs, grasslands, peats, agricultural residues, and the burning of biofuels in the residential sector (stoves, boilers, fireplaces). We removed the wildfire component, resulting in anthropogenic biomass emissions (such as those from agricultural residues and dung cakes, which we assume are reported in NGHGIs) and biofuel burning (corresponding to IPCC category 1.A.4 in NGHGIs). Consequently, the scope of our methane inversion flux,

after removing wildfires, is broader than that of wood fuel emissions. This is likely a primary reason for the differences observed.

We have included this comparison and related analysis in the revised manuscript and supplementary materials:

**Line 314-316**:

*This method has some uncertainties. First, the partitioning relies on prior fractions within each pixel, and second, emissions from wildfires are counted for in the Biomass and Biofuel burning (BB) inversion category while they are not necessarily reported in NGHGIs. The BB inversion category includes methane emissions from wildfires in forests, savannahs, grasslands, peats, agricultural residues, and the burning of biofuels in the residential sector (stoves, boilers, fireplaces). Therefore, we subtracted bottom-up (BU) emissions from wildfires ($E_{wildfires}^{BU}$) based on the GFEDv4 dataset (van Wees et al., 2022) using their reported dry matter burned and $CH_4$ emission factors. Because the GFEDv4 dataset also reports specific agricultural and waste fire emissions data, we assumed that those fires (on managed lands) are reported by NGHGIs, so they were not counted in $E_{wildfires}^{BU}$.* ***Figure S3** presents a comparison between our adjusted BB flux and the wood fuel emissions reported by Flammi et al. (2023). This comparison highlights the broader scope and definition of our adjusted BB flux, illustrating the differences in emissions estimation methodologies.*

**Fig S3** in supplementary materials:

[Figure]

*Fig S3. Comparison of the adjusted CH4 inversion flux from the biomass and biofuel burning sector (after excluding wildfires in this study, G: in-situ inversion ensembles; S: satellite-based inversion ensembles) with the bottom-up calculated wood fuel emissions from Flammi et al. (2023) (F). The analysis includes eight countries: China (CHN), India (IND), Iran (IRN), Brazil (BRA), Argentina (ARG), Venezuela (VEN), Nigeria (NGA), and Mexico (MEX). The wood fuel emissions reported by Flammi et al. are generally lower than our estimated emissions. This discrepancy can be attributed to the broader scope of our methane inversion results. Flammi et al. estimate wood fuel emissions for 125*

*countries using the UNSD Energy Statistics database, focusing specifically on the non-renewable biomass fraction utilized in cooking. In contrast, our study encompasses a wider range of emissions, including anthropogenic sources such as agricultural residues and dung cakes, while excluding wildfires. This methodological difference likely accounts for the observed variations in emissions estimates.*

4. Lateral carbon transport by crop and wood products- This lateral carbon transport is a really interesting aspect of your work. However, could you highlight this aspect more in the results section? Could you perhaps discuss the extent to which these emissions/uptakes affect total emissions/uptake. Also are these just based on primary product trade (e.g. roundwood) or do they include primary and secondary trade (e.g. roundwood would be primary but wood pulp, sawn wood would be secondary)

We appreciate your interest in the aspect of lateral carbon transport. We have expanded the discussion in the results section to emphasize its impact:

**Line 450-455**: *In these countries, adjusting inversions by CO2 fluxes induced by river carbon transport and by the trade of crop and wood products tends to lower CO2 sinks, especially for large crop exporters like the USA and Canada. The adjusted net lateral transport fluxes for these countries are 48 (China), 143 (USA), 86 (EU), 63 (Russia), 72 (Canada), 75 (India), and 145 (Brazil) TgC/yr, which represent 20%, 38%, 48%, 11%, 41%, 94%, and 60% of the managed land CO2 fluxes before lateral transport adjustments, respectively.*

Regarding wood products, we clarified in Line 216 that we utilized the bookkeeping model results from Ciais et al. (2021). According to Ciais et al., all direct and indirect products are considered. Specifically, "For roundwood (FAO code 1861), FAOSTAT data were used, and for processed products potentially entering international trade, the GTAP-MRIO data were employed."

We included this explanation in the main text to clarify that both primary and secondary trade (including roundwood, wood pulp, and sawn wood) is accounted for:

**Lines 216-220**: *Here, we followed Ciais et al. (2021), who used a bookkeeping model to calculate the fraction of domestically produced and imported carbon in wood products that are oxidized in each country during subsequent years, with product lifetimes defined by Mason Earles et al. (2012) and encompassing all products (including roundwood and processed products).*

**References**:
Ciais, P., Yao, Y., Gasser, T., Baccini, A., Wang, Y., Lauerwald, R., Peng, S., Bastos, A., Li, W., Raymond, P. A., Canadell, J. G., Peters, G. P., Andres, R. J., Chang, J., Yue, C., Dolman, A. J., Haverd, V., Hartmann, J., Laruelle, G., Konings, A. G., King, A. W., Liu, Y., Luyssaert, S., Maignan, F., Patra, P. K., Peregon, A., Regnier, P., Pongratz, J., Poulter, B., Shvidenko, A., Valentini, R., Wang, R., Broquet, G., Yin, Y., Zscheischler, J., Guenet, B., Goll, D. S., Ballantyne, A.-P., Yang, H., Qiu, C., and Zhu, D.: Empirical estimates of

regional carbon budgets imply reduced global soil heterotrophic respiration, Natl Sci Rev, 8, nwaa145, 2021.

5. Heatmap of countries selected for inversion data- Based on the discussion on lines 99-101 on page 3, it would be interesting to see the countries selected as a heatmap just to understand what portion of emissions are covered globally by emissions type.

Thank you for the comment. We've added **Fig S1** to illustrate the portion of emissions by each gas:

[Figure]

*Fig S1. Portion of selected countries' emissions to global fossil CO2 emissions (green blocks), anthropogenic CH4 emissions (blue blocks), and anthropogenic N2O emissions (red blocks).*

And we add a citation in **Line 102**:

*According to the median of inversion data we used in this study, selected countries collectively represent ~70% of global fossil fuel CO2 emissions, ~90% of global land CO2 sink, ～60% of anthropogenic CH4 emissions, and ~55% of anthropogenic N2O emissions (Fig S1).*

Other minor comments-

1. Lines 92 on Page 3, it seems that there is a typo " Atmosphericnversions" should be separate words.

Thanks, we've revised it accordingly.

2. Line 170 on Page 7- What do advection and convection schemes mean? Can an explanation be added in a footnote or maybe even be explained in text?

We appreciate your suggestion for clarification. We have added an explanation of "advection" and "convection schemes" in the main text:

**Line 174-177**: *This ensemble of inversions gathers various chemistry transport models, differing in vertical and horizontal resolutions, meteorological forcing, advection (horizontal transport of air due to wind) and convection (vertical transport) schemes, and boundary layer mixing (detailed characteristics can be found in Table S11 in Saunois et al. 2024).*

3. Line 176 on Page 7- Is there a reference missing for "GAINS"?

We've added the reference to GAINS and completed the sentence:
**Line 180-182**: *During the production of the inversion simulations, GAINS inventory (Höglund-Isaksson, 2020) was proposed to use another prior for fossil fuel sources instead of using EDGAR v6.*

4. Lines 290-293 on Page 11. " *However, the differences in the calculated results among the four methods were smaller compared to the variations observed in the inversions (see Deng et al. (2022) Fig 9).*" I think you need a sentence after this just summarizing what the differences are before you explain the method used.

We've revised the section to include a concise summary of the differences among the four methodologies, and illustrated that "the differences" would actually be the "uncertainty from the separation method".

**Line 295-305**: *In our previous study (Deng et al., 2022), four methods were proposed to separate CH₄ anthropogenic emissions from inversions ($E_{Anth}^{inv}$) to compare them with national inventories ($E_{Anth}^{ni}$) aiming to discuss the uncertainties in anthropogenic CH₄ emissions associated with the chosen separation methods. These four methods include: (1) summing prior estimates based on inversions for anthropogenic sectors (method 1); (2) subtracting natural emissions from total fluxes (method 2); and (3) subtracting natural emissions derived from other bottom-up assessments from the total inversion flux (methods 3/1 and 3/2, differing only in the bottom-up wetland CH₄ data used). The calculations of anthropogenic emissions by each method were performed separately for GOSAT inversions and in-situ inversions. However, the uncertainty from the separation method is generally much smaller than the variability between different inversion models (see Deng et al. (2022) Fig 9). Therefore, we apply only one method in this study which consists of using inversion partitioning as defined in Saunois et al. (2020):*

5. Lines 443-444 on Page 17- Formatting is off for this sentence- "*post fire biomass changes suggest that fire emissions have exceeded regrowth on average in Western Canada and Alaska until ≈ 2010*"

Thanks, we've corrected the sentence accordingly.

---

## Author Response (AR2)

Public justification (visible to the public if the article is accepted and published):
Please make the following changes:

line 255. When citing FAOSTAT data, please add a citation. The citation should have the follwing format: FAO, 2024. FAOSTAT "Name of exact dataset", available at : "link to exact dataset (not generic FAOSTAT page)". FAO, Rome, Italy. Downloaded on "date"

Thanks. We've added the citation.

Line 255: " *(Xu et al., 2021; FAO, 2024).* "
Line 1065: "*FAO: Trade, FAOSTAT, 2024. available at: https://www.fao.org/faostat/en/#data. FAO, Rome, Italy.*"

line 315. Please correct, exact citation is "Flammini et al.", not "Flammi et al."

Corrected.

Line 314: "*reported by Flammini et al. (2023).*"

Furthermore, please consider (not required for publication) adjusting communication of uncertainty and associated numerics throughout the manuscript, in a manner that better aligns to the rules of inferential statistics.

Examples:

line 450. How can 210±180 be referred to a "median sink"? The statistical inference cannot be used to state this flux is either a sink or a source.
line 463. I am not sure how a precision of 45 can be used to infer a mean/median value of 125, as in 125±45. Perhaps writing this result as 130±50 would be more appropriate
line 522. For a country like DOC, where a lot of uncertainty in underlining land based statistics is likely very high, is it credible that precision in CO2 flux is as high as in the stated results, for instance 135±0.1 or 95±0.5
653 It is not good practice to have errors that are more precise that the inferenced mean, as in 20±1.6. This result should rather be expressed as 20±2 (and 26±1.6 should be rather communicated as 26±2)

Thanks for the suggestion. We added a description in Line 445 to explain how the reported values and uncertainty intervals are presented:
*"In this paper, for inversion results covering a time interval, we present the data as mean ± standard deviation, where the mean is the multi-year average of the median flux values from the inversion models, and the standard deviation represents the interannual variability.*
*"*

Specifically, for 210 ± 180 (Line 450), it represents the 2010s average of the median flux values across five in-situ $CO_2$ models, where 210 is the multi-year average and 180 denotes the interannual variability. We clarified that the median flux values indicate a net carbon sink, despite the large uncertainty range.

We also reviewed and adjusted the numerics throughout the manuscript to better align with the rules of inferential statistics.

Line 463: "*Like in Deng et al. 2022, we found that the carbon sink of Canada's managed land is significantly larger (-130 ± 50 TgC/yr over 2001-2021 from in-situ inversions) than the NGHGIs reports (5 ± 4 TgC/yr over 2001-2021).* "

Line 522: "Since 2000, the NGHGIs reports indicated three stages of different levels of CO2 flux, which COD managed land was a carbon source during 2000-2010 (~95 TgC/yr), a larger carbon source during 2011-2014 (~135 TgC/yr), and a very small sink during 2015-2018 (~-1 TgC/yr). "

Line 653: "*In CHN, the in-situ (20 ± 2 Tg CH4/yr) and satellite inversions (17 ± 1 Tg CH4/yr) emissions in the 2010s are 24% and 35% lower than in the NGHGIs (~26 ± 2 Tg CH4/yr), respectively.*"